# Supporting Multimodal Intermediate Fusion with Informatic Constraint and Distribution Coherence

**Yi Li**[1,2,*], **Fei Song**[1,2,*], **Changwen Zheng**[1] & **Jiangmeng Li**[1,†]
1. Institute of Software Chinese Academy of Sciences
2. University of Chinese Academy of Sciences
{liyi2022,songfei2022,changwen,jiangmeng2019}@iscas.ac.cn

## Abstract

Based on the prevalent intermediate fusion (IF) and late fusion (LF) frameworks, multimodal representation learning (MML) demonstrates its superiority over unimodal representation learning. To investigate the intrinsic factors underlying the empirical success of MML, research grounded in theoretical justifications from the perspective of generalization error has emerged. However, these provable MML studies derive the theoretical findings based on LF, while theoretical exploration based on IF remains scarce. This naturally gives rise to a question: ***Can we design a comprehensive MML approach supported by the sufficient theoretical analysis across fusion types?*** To this end, we revisit the IF and LF paradigms from a fine-grained dimensional perspective. The derived theoretical evidence sufficiently establishes the superiority of IF over LF under a specific constraint. Based on a general $K$-Lipschitz continuity assumption, we derive the generalization error upper bound of the IF-based methods, indicating that eliminating the distribution incoherence can improve the generalizability of IF-based MML methods. Building upon these theoretical insights, we establish a novel IF-based MML method, which introduces the informatic constraint and performs distribution cohering. Extensive experimental results on multiple widely adopted datasets verify the effectiveness of the proposed method.

## 1 Introduction

Given the gradually increasing data from multiple modalities, multimodal representation learning (MML) demonstrates the potential for supporting the comprehension of complex patterns. According to the feature mapping stages Wang et al. (2020), two widely adopted multimodal fusion types exist in recent MML studies, i.e., feature-level *intermediate* fusion (IF) and decision-level *late* fusion (LF) [1]. IF integrates features from various modalities in the latent space, whereas LF merges the prediction logits in the target space. MML has recently arisen as a popular area of research in many fields, e.g., knowledge graph Cao et al. (2022); Lu et al. (2022), recommendation Zhou et al. (2023); Wei et al. (2023); Li et al. (2024), sentiment analysis Hazarika et al. (2020); Li et al. (2023); Liu et al. (2024), and so on. Besides the documented empirical success, studies Zhang et al. (2023b); Cao et al. (2024) investigate the inherent mechanisms behind MML from the generalization error perspective, thereby providing theoretical supports for the multimodal models.

However, thus-far provable works from the generalization error perspective derive theorems based on the LF framework, while the theoretical analysis focusing on the IF framework remains insufficiently explored. Theoretically, according to the theory of data processing inequality Cover & Thomas (2001), IF-based methods may contain more task-dependent information. Empirically, we conduct exploratory experiments by substituting the framework of two representative LF-based methods (PDF Cao et al. (2024) and QMF Zhang et al. (2023b)) with IF. As illustrated in Figure 1,

---

[*] Equal contribution. [†] Corresponding author.

[1]Early fusion aggregates the original data directly, which is impractical in real-world scenarios due to the heterogeneity of multimodal data. Therefore, we remove early fusion from the consideration.

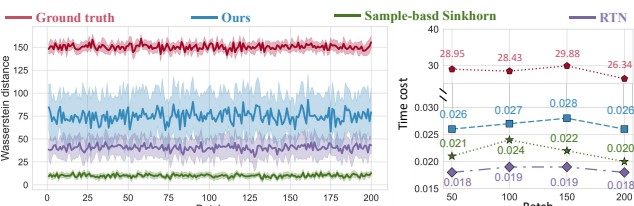

Figure 1: QMF (IF) replaces the LF framework in QMF with the IF, and the same applies to PDF (IF). MVSA-Single, MVSA-Multiple, HFM, and Food101 are four vision-language datasets.

IF-based methods consistently outperform their LF-based counterparts on four multimodal datasets. Despite the theoretical and empirical potentials of IF's ascendancy over LF, the theoretical supports behind IF-based MML models require further exploration. To this end, we revisit the IF and LF paradigms from a fine-grained dimensional perspective. With rigorous deduction, we demonstrate the superiority of IF over LF under a specific constraint. Therefore, we design our model based on the IF framework and incorporate a specific informatic constraint. The informatic constraint imposes a regularization on parameters of the linear target mapping in IF-based MML models from the information theory perspective Tishby et al. (2000). Such an informatic constraint can sufficiently guarantee the superiority of IF over LF.

To further explore the inherent mechanism behind IF-based MML models, we formalize the generalization error upper bound of IF-based methods, which is derived by adhering to a general $K$-Lipschitz continuity assumption on the linear target mapping. Observing the generalization error upper bound, we reveal that eliminating the distribution incoherence can improve the generalization performance of IF-based MML models. Thus, we determine to employ Wasserstein distance to conduct distribution cohering for its favorable properties. Directly calculating Wasserstein distance Cuturi (2013) between high-dimensional features requires huge computational complexity. Accord-

Figure 2: On the MVSA-Single dataset, we leverage three methods to estimate Wasserstein distance and record the average time cost per 50 batches. The ground truth Wasserstein distance is obtained by applying the Sinkhorn algorithm to all high-dimensional features in each batch. The results show that directly computing Wasserstein distance is impracticable because of the high time complexity. From the left column of the figure, it can be observed that compared to sampling-based Sinkhorn and RTN, our method achieves a more accurate Wasserstein distance estimation with a limited increase in time complexity.

ingly, two main categories of methods are proposed to practically estimate Wasserstein distance: (i) Sampling-based Sinkhorn Cao et al. (2022); Li et al. (2023); (ii) Radon transform-based nonlinear neural network calculation (RTN) Bonneel et al. (2015); Kolouri et al. (2019); Chen et al. (2022); Sugimoto et al. (2024). Nevertheless, due to the incompleteness of the partial sampling strategy in sampling-based Sinkhorn and the inaccuracy of fitted non-linear functions in RTN, current methods suffer from the degraded estimation of Wasserstein distance, as demonstrated in Figure 2. To address this issue, we propose a novel estimation method of Wasserstein distance, which introduces a restricted isometric dimensionality reduction technique, and design a Lagrange regularization to enhance robustness to the semantic disturbance during dimensionality reduction. This approach empowers us to omit the partial sampling strategy and the nonlinear neural network, thus achieving distribution cohering effectively with limited computational complexity. The empirical evidence in Figure 2 verifies our statement.

In a nutshell, we propose a novel IF-based MML method with solid theoretical supports, namely *Intermediate Fusion with Informatic Constraint and Distribution Coherence* (**IID**). Our major contribution is four-fold:

**(1)** From a fine-grained dimensional perspective, we rethink the two prevalent fusion types of MML, i.e., IF and LF. We theoretically demonstrate the superiority of IF-based methods over LF-based counterparts based on a specific constraint. **(2)** Based on the $K$-Lipschitz continuity assumption on the linear target mapping, we derive the generalization error upper bound of IF-based methods,

which indicates that mitigating the distribution incoherence can improve the generalizability of IF-based MML models. **(3)** Adhering to theoretical analyses, we propose a novel IF-based MML model, encompassing informatic linear target mapping constraint and distribution cohering with restricted isometric dimensionality reduction. **(4)** Empirically, we conduct extensive experiments on representative benchmarks to prove the effectiveness of IID.

## 2  RELATED WORK

In recent years, the expansion of available data has significantly propelled advancements in the fields of computer vision Krizhevsky et al. (2012); He et al. (2016); Huang et al. (2017); Dosovitskiy et al. (2021) and natural language processing Pennington et al. (2014a); Vaswani et al. (2017); Devlin et al. (2019b), enabling the development of more robust and sophisticated applications. However, these models focus on the processing of unimodal data (e.g., images and text). As the semantics extracted from unimodal data approach its bottleneck, MML has garnered increasing attention from the research community. By exploring both the modality-shared and modality-specific task-dependent discriminative knowledge, MML demonstrates its superiority in fields involving various modality combinations, like audio-video-text Liu et al. (2024); Hazarika et al. (2020), image-texts Li et al. (2023); Ma et al. (2024), graph-image-texts Wei et al. (2023); Cao et al. (2022), and so on.

Besides the documented empirical success, research endeavoring to understand MML with theoretical justifications has started to emerge. E.g., Huang et al. (2021) rigorously demonstrates that the reason why MML outperforms unimodal methods lies in its access to a superior latent space representation. Huang et al. (2022) substantiates the existence of modality competition, which renders the joint training of multimodal networks challenging, thereby leading to suboptimal performance. Beyond the exploration of the intrinsic mechanism of MML, several works develop multimodal models under the theoretical guidance of generalization error and yield great success. Specifically, QMF Zhang et al. (2023b) is designed by the theoretical derivation that the negative correlation between a specific modality's fusion weight and empirical error can decrease the generalization error. PDF Cao et al. (2024) is proposed based on the provable elucidation that the reduction of generalization error primarily stems from the negative covariance between fusion weights and the loss associated with the current modality, as well as the positive covariance between fusion weights and the loss of other modalities. Due to the inherent correspondence between the ensemble-like LF framework and the extensively investigated field of ensemble learning Qiao & Peng (2024); Wood et al. (2023), these works consistently derive the theoretical findings based on the LF framework, thus resulting in the sparse theoretical exploration based on IF. In contrast to prior research, we design a comprehensive MML approach, supported by a complete theoretical analysis across fusion types.

## 3  THEORETICAL INSIGHTS

This section presents our theoretical insights, and we offer a concise overview of the proposed theorems with complete proofs deferred to **Appendix** A.2.

We first provide the basic notations of MML. We denote the input space, latent space, and target space by $\mathcal{X}$, $\mathcal{Z}$, and $\mathcal{Y}$, respectively. Given a multimodal learning task, the training dataset $\mathcal{D}_{\text{train}}$ comprises instances of the form $(\boldsymbol{x}, y)$, which are sampled from the distribution $\mathcal{D} \in \mathcal{X} \times \mathcal{Y}$. $\boldsymbol{x}$ is the multimodal sample and $y$ is the corresponding label. Two mappings are defined to assist our theoretical analysis: ($\boldsymbol{i}$) latent mapping $h(\cdot) : \mathcal{X} \mapsto \mathcal{Z}$, which takes an input from the input space $\mathcal{X}$ and projects it into the latent space $\mathcal{Z}$; ($\boldsymbol{ii}$) target mapping $g(\cdot) : \mathcal{Z} \mapsto \mathcal{Y}$, which takes latent features from the latent space $\mathcal{Z}$ and maps them to the target space $\mathcal{Y}$. The formula $f = g \circ h(\boldsymbol{x})$, abbreviated as $f = gh(\boldsymbol{x})$, is a composite function of $g(\cdot)$ and $h(\cdot)$. Our objective is to learn a multimodal model $f$ that performs well on the unknown test dataset $\mathcal{D}_{\text{test}}$, which is also drawn from $\mathcal{D}$.

### 3.1  REVISITING THE IF AND LF PARADIGMS: A FINE-GRAINED DIMENSIONAL PERSPECTIVE

For the sake of simplicity and without loss of generality, we perform the theoretical analysis within the scenario involving two modalities. Given the input multimodal data $\boldsymbol{x} = \{x_1, x_2\}$, we employ latent mappings to obtain the corresponding features by $\boldsymbol{z}_1 = h^1(x_1)$ and $\boldsymbol{z}_2 = h^2(x_2)$. Given the $m$-th ($m \in \{1, 2\}$) modality-specific fusion weight $w^m > 0$ and $\sum_{m=1}^{2} w^m = 1$, for LF, the final

prediction logits $f_{\text{LF}}(\boldsymbol{x}) = \sum_{m=1}^{2} w^m g_{\boldsymbol{\theta}_m}^m h^m(x_m)$, while for IF, $f_{\text{IF}}(\boldsymbol{x}) = g_{\boldsymbol{\theta}}[\sum_{m=1}^{2} w^m h^m(x_m)]$. It can be seen that each modality has its specific target mapping $g^m(\cdot)$ parameterized by $\boldsymbol{\theta}_m$ in LF. In contrast, IF leverages a common target mapping $g(\cdot)$ parameterized by $\boldsymbol{\theta}$ for multiple modalities. Being consistent with our major baseline Zhang et al. (2023a); Cao et al. (2024), we employ a linear classification layer as our target mapping, which is a widely adopted setting in multimodal learning tasks Anderson et al. (2018); Han et al. (2021); Cao et al. (2024).

We assume that $\boldsymbol{z}_1$ and $\boldsymbol{z}_2$ share the same dimension $\mathbb{R}^d$, which can be realized easily by a linear transform in practice. Intuitively, in the image classification task, a specific pixel of a picture either belongs to the task-dependent foreground or to the task-irrelevant background. Analogously, from a fine-grained perspective, each dimension of latent features $\boldsymbol{z}_1$, $\boldsymbol{z}_2$ (e.g., $z_{1,n}$ and $z_{2,n}, 1 \leq n \leq d$) is either task-dependent semantics or task-independent noise. We provide the definition of task-dependent semantic and task-independent noisy dimensions.

**Definition 1 (Semantic and noisy dimensions).** *If masking a given dimension results in a decrease of the error between the model's predictions and the ground truth label, the dimension is classified as a task-dependent semantic dimension; conversely, the dimension is classified as a task-independent noisy dimension.*

Thus, there are two partitions corresponding to per latent feature, i.e., $\boldsymbol{z}_1 = \{\boldsymbol{z}_{1,S_1}, \boldsymbol{z}_{1,N_1}\}, \boldsymbol{z}_2 = \{\boldsymbol{z}_{2,S_2}, \boldsymbol{z}_{2,N_2}\}$, where $S_m$ and $N_m$ denote the index sets of semantic dimensions and noisy dimensions, respectively, corresponding to the $m$-th modality. $S_1 \cap N_1 = \oslash$ and $S_2 \cap N_2 = \oslash$ since a certain dimension cannot be semantics and noise simultaneously. In LF, the parameters $(\boldsymbol{\theta}_1, \boldsymbol{\theta}_2)$ of the target mappings also have two partitions corresponding to the input latent features, i.e., $\boldsymbol{\theta}_1 = \{\boldsymbol{\theta}_{1,S_1}, \boldsymbol{\theta}_{1,N_1}\}, \boldsymbol{\theta}_2 = \{\boldsymbol{\theta}_{2,S_2}, \boldsymbol{\theta}_{2,N_2}\}$. Then the prediction logits of LF can be formalized as

$$f_{\text{LF}}(\boldsymbol{x}) = w^1(\boldsymbol{z}_1 \boldsymbol{\theta}_1) + w^2(\boldsymbol{z}_2 \boldsymbol{\theta}_2) = w^1 \boldsymbol{z}_{1,S_1} \boldsymbol{\theta}_{1,S_1} + w^1 \boldsymbol{z}_{1,N_1} \boldsymbol{\theta}_{1,N_1} + w^2 \boldsymbol{z}_{2,S_2} \boldsymbol{\theta}_{2,S_2} + w^2 \boldsymbol{z}_{2,N_2} \boldsymbol{\theta}_{2,N_2}. \tag{1}$$

While in IF, the multimodal feature is obtained in latent space by $\boldsymbol{z} = w^1 \boldsymbol{z}_1 + w^2 \boldsymbol{z}_2$, thus each dimension of $\boldsymbol{z}$ has four possible scenarios:

- $\mathbb{D}_{S_1 S_2} = S_1 \cap S_2$, a combination of the semantic dimensions of $\boldsymbol{z}_1$ and $\boldsymbol{z}_2$;

- $\mathbb{D}_{S_1 N_2} = S_1 \cap N_2$, a combination of the semantic dimension of $\boldsymbol{z}_1$ and the noisy dimension of $\boldsymbol{z}_2$;

- $\mathbb{D}_{N_1 S_2} = N_1 \cap S_2$, a combination of the noisy dimension of $\boldsymbol{z}_1$ and the semantic dimension of $\boldsymbol{z}_2$;

- $\mathbb{D}_{N_1 N_2} = N_1 \cap N_2$, a combination of the noisy dimensions of $\boldsymbol{z}_1$ and $\boldsymbol{z}_2$.

Briefly, $\boldsymbol{z}$ can be partitioned into four components $\{\boldsymbol{z}_{\mathbb{D}_{S_1 S_2}}, \boldsymbol{z}_{\mathbb{D}_{S_1 N_2}}, \boldsymbol{z}_{\mathbb{D}_{N_1 S_2}}, \boldsymbol{z}_{\mathbb{D}_{N_1 N_2}}\}$, and arbitrary two sets in $\{\mathbb{D}_{S_1 S_2}, \mathbb{D}_{S_1 N_2}, \mathbb{D}_{N_1 S_2}, \mathbb{D}_{N_1 N_2}\}$ are disjoint obviously. Corresponding to the fused multimodal feature $\boldsymbol{z}$, the parameter $\boldsymbol{\theta}$ of the target mapping has four partitions, i.e., $\boldsymbol{\theta} = \{\boldsymbol{\theta}_{\mathbb{D}_{S_1 S_2}}, \boldsymbol{\theta}_{\mathbb{D}_{S_1 N_2}}, \boldsymbol{\theta}_{\mathbb{D}_{N_1 S_2}}, \boldsymbol{\theta}_{\mathbb{D}_{N_1 N_2}}\}$. Accordingly, the prediction logits of IF is

$$f_{\text{IF}}(\boldsymbol{x}) = \boldsymbol{z} \cdot \boldsymbol{\theta} = \boldsymbol{z}_{\mathbb{D}_{S_1 S_2}} \boldsymbol{\theta}_{\mathbb{D}_{S_1 S_2}} + \boldsymbol{z}_{\mathbb{D}_{S_1 N_2}} \boldsymbol{\theta}_{\mathbb{D}_{S_1 N_2}} + \boldsymbol{z}_{\mathbb{D}_{N_1 S_2}} \boldsymbol{\theta}_{\mathbb{D}_{N_1 S_2}} + \boldsymbol{z}_{\mathbb{D}_{N_1 N_2}} \boldsymbol{\theta}_{\mathbb{D}_{N_1 N_2}}. \tag{2}$$

Then, we can derive the following Theorem 1.

**Theorem 1 (Prediction comparisons of IF and LF).** *For each input multimodal sample $(\boldsymbol{x}, y)$, there constantly exists a set of parameters $\Lambda$, such that the following equation holds for the linear target mapping characterized by $\boldsymbol{\theta} \in \Lambda$:*

$$\mathcal{L}(f_{\boldsymbol{\theta},IF}(\boldsymbol{x}), y) \leq \mathcal{L}(f_{LF}(\boldsymbol{x}), y), \tag{3}$$

*where $\mathcal{L}(\cdot, \cdot)$ is Cross-Entropy loss function. Given the Bayes optimal hypothesis $f^*$, which achieves the infimum of the errors $\mathcal{R}^*$ on $\mathcal{D}$, i.e., $f^* = argmin_f \mathcal{R}(f) = argmin_f \mathbb{E}_{(\boldsymbol{x},y) \sim \mathcal{D}}[\mathcal{L}(f(\boldsymbol{x}), y)]$, and for each $\epsilon \in \left[0, \| \mathcal{L}(f^*(\boldsymbol{x}), y) - \mathcal{L}(f_{LF}(\boldsymbol{x}), y) \| \right]$, there exists a corresponding $\boldsymbol{\theta}' \in \Lambda$ s.t. $\mathcal{L}(f_{\boldsymbol{\theta}',IF}(\boldsymbol{x}), y) = \mathcal{L}(f_{LF}(\boldsymbol{x}), y) - \epsilon$.*

The proof of Theorem 1 can be found in **Appendix** A.2.1. We omit the explicit notation of target mappings' parameters in LF-based prediction, since Theorem 1 holds for arbitrary parameters of target mappings in LF models (This paper follows this notation principle throughout). Theorem 1 confirms that a simple linear target mapping characterized by the parameters in $\Lambda$ can establish the superiority of IF over LF. Additionally, there theoretically exists a $\boldsymbol{\theta}'$ that allows the IF-based prediction to be closer to Bayesian optimal prediction compared to those of the LF models.

## 3.2 ANALYSIS OF THE GENERALIZATION ERROR

Based on Theorem 1, we present our theorem regarding the generalization error of IF and LF. The generalization error is a metric that measures the generalization performance of the learned multi-modal model $f$, which can be defined as: $\mathscr{G} = \mathbb{E}_{(\boldsymbol{x},y)\sim\mathcal{D}}[\mathcal{L}(f(\boldsymbol{x}),y)]$. Theorem 2 delineates the comparison of generalization errors between IF and LF.

**Theorem 2 (Generalization errors of IF and LF)**. *With a linear target mapping $g_{\boldsymbol{\theta}}$ in IF parameterized by $\boldsymbol{\theta} \in \Lambda$, the following equation holds: $\mathscr{G}_{IF,\boldsymbol{\theta}} \leq \mathscr{G}_{LF}$.*

Theorem 2 is proven in **Appendix** A.2.2, which indicates that IF with linear target mapping $g_{\boldsymbol{\theta}}(\cdot)$ can exhibit lower generalization error than LF consistently. We further introduce Assumption 1 to investigate the factors impacting the generalization error of IF-based MML methods.

**Assumption 1 ($K$-Lipschitz continuity)**. *Suppose the function $\phi(\boldsymbol{z}) = \mathcal{L}(g(\boldsymbol{z}),y)$ is a $K$-Lipschitz continuous function in respect to input $\boldsymbol{z}$, then for $K > 0$, $\forall~a,b \in \mathcal{D}_\phi$ ($\mathcal{D}_\phi$ is the definitional domain of $\phi$), we have: $\|~\phi(a) - \phi(b)~\| \leq K~\|~a-b~\|$.*

Analogous assumption has also been adopted in Qiao et al. (2025), various existing works Arjovsky & Bottou (2017); Arjovsky et al. (2017); Cao et al. (2022) introduce the constraint of $K$-Lipschitz continuity assumption within their theoretical analysis, demonstrating the generality of $K$-Lipschitz continuity constraint. Furthermore, relevant studies Yoshida & Miyato (2017); Gulrajani et al. (2017) declare that a $K$-Lipschitz continuous function can be easily constructed. The literature indicates that $K$-Lipschitz continuity constitutes a mild assumption.

**Theorem 3 (Generalization error upper bound of IF)**. *Let $\mathcal{D}_{train} = \{\boldsymbol{x}^i, y^i\}_{i=1}^{|\mathcal{D}_{train}|}$ be the training dataset and $\mathcal{D}_\mathcal{M}$ be a complete distribution distance metric. Under the constraint condition of Assumption 1, for any $f_{IF}$ with the linear target mapping $g_{\boldsymbol{\theta}}$ parameterized by $\boldsymbol{\theta} \in \Lambda$ in hypothesis space $\mathcal{H}$ and $0 < \delta < 1$, with the probability at least $1-\delta$, the generalization error of $f_{IF}$ holds:*

$$\mathscr{G}_{IF,\boldsymbol{\theta}} \leq \sum_{m=1}^{M}\left[ K\cdot\mathbb{E}(w^m)\underbrace{\mathcal{D}_\mathcal{M}(\mu_m,\mu)}_{\text{Distribution incoherence}} + \text{Error}\left(w^m, \mathcal{L}(g_{\boldsymbol{\theta}}(\boldsymbol{z_m}),y)\right)\right] + \hat{\mathbb{E}}(f_{IF}) + \text{Bias}\left[\mathfrak{R}(\mathcal{H}), \mathcal{O}(N^{-1/2})\right].$$

$$(4)$$

The corresponding proof of Theorem 3 can be found in **Appendix** A.2.3. $\mathbb{E}(w^m)$ represents the expectation of multimodal fusion weight, $\mathcal{D}_\mathcal{M}$ is the complete distribution distance metric which satisfies the three essential properties (non-negativity, symmetry, triangle inequality). $\mu_m$ is the distribution that the features of the $m$-th modality are drawn from, $\mu$ is the distribution that the multimodal feature $\boldsymbol{z}$ follows. Distribution incoherence quantifies the discrepancy between the distributions $\mu$ and $\mu_m$ ($m \in [1, M]$). $\hat{\mathbb{E}}(f_{\text{IF}})$ is the empirical error of multimodal feature $\boldsymbol{z}$ on $\mathcal{D}_{\text{train}}$. $\text{Bias}[\mathfrak{R}(\mathcal{H}), \mathcal{O}(N^{-1/2})]$ is the systematic bias with respect to Rademacher complexity $\mathfrak{R}$ of the hypothesis space $\mathcal{H}$ and the size of training dataset $N$. It's challenging to eliminate the systematic bias in MML models. $\text{Error}[w^m, \mathcal{L}(g_{\boldsymbol{\theta}}(\boldsymbol{z_m}),y)]$ is an error term about the fusion weight $w^m$ and unimodal loss $\mathcal{L}(g_{\boldsymbol{\theta}}(\boldsymbol{z_m}),y)$, which indicates that the calculation method of fusion weight $w^m$ can affect the predictive performance of MML models. Recent research Zhang et al. (2023a); Cao et al. (2024) focuses on exploring the effective fusion weights $w^m$ to achieve better performance of MML models, leaving the diminution of the distribution incoherence term unexplored.

Consequently, inspired by Theorems 1 and 2, we determine to implement our model based on the IF framework with a linear target mapping characterized by $\boldsymbol{\theta} \in \Lambda$. According to Eq.(4) in Theorem 3, we propose to diminish the value of the distribution incoherence term, thereby further enhancing the generalizability of our method.

## 4 METHODOLOGY

**Overview of IID.** The framework of the proposed IID is illustrated in Figure 3. Drawing upon Theorems 1 and 2, we build our model based on the IF framework. Concretely, given a batch of input multimodal samples $\{(\boldsymbol{x}^1,y^1),(\boldsymbol{x}^2,y^2),\cdots,(\boldsymbol{x}^N,y^N)\}$, $N$ is the batch size, each instance has $M$ modalities, i.e., $\boldsymbol{x}^i = \{x_1^i, x_2^i, \cdots, x_M^i\}$ ($i \in [1, N]$), and we obtain the corresponding features by

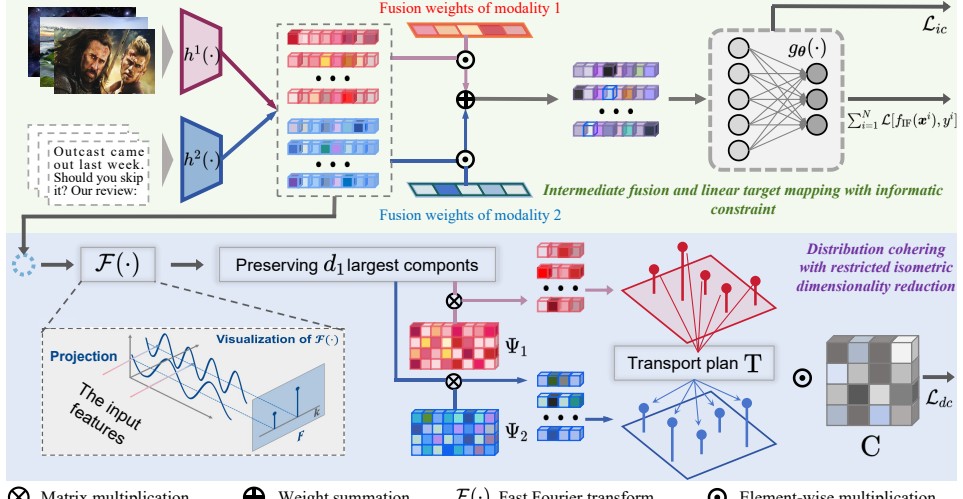

Figure 3: The overall architecture of IID, which is built based on a prevalent IF framework. The pipeline is illustrated under the scenario of two modalities without loss of generality. The proposed informatic constraint on linear target mapping and distribution cohering with restricted isometric dimensionality reduction bridges our theoretical framework and practical methodology seamlessly.

latent mappings, i.e., $z_m^i = h^m(x_m^i)$, where $m \in [1, M]$. Then we obtain the multimodal feature $z^i$ of $x^i$ in latent space via

$$z^i = \sum_{m=1}^{M} w^m z_m^i, \tag{5}$$

which is a prevalent IF paradigm, and we calculate the prediction logits by $f_{\mathrm{IF}}(x^i) = g(z^i)$.

The derivations of Theorems 2 and 3 are based on the linear target mapping parameterized by $\theta \in \Lambda$. To actualize such a specific and accessible linear target mapping, we introduce the meticulously designed informatic constraint, which guarantees that the parameter of the linear target mapping is restricted to the desired set $\Lambda$ and converges towards the theoretically optimal parameter $\theta^*$ during the training process. Under the guidance of Theorem 3, we propose the distribution cohering with restricted isometric dimensionality reduction module to diminish the distribution incoherence term in Eq.(4), thereby improving the generalizability of the proposed IID.

## 4.1 Linear target mapping with informatic constraint

In this section, we introduce the informatic constraint to attain the expected linear target mapping. As delineated in Theorem 1, based on $\theta \in \Lambda$, we have $\mathcal{L}(f_{\mathrm{IF},\theta}(x^i), y^i) \le \mathcal{L}(f_{\mathrm{LF}}(x^i), y^i)$, which equals $\mathcal{L}(z^i \cdot \theta, y^i) \le \sum_{m=1}^{M} w^m \mathcal{L}(z_m^i \cdot \theta_m, y^i)$. Therefore, given the initial parameter $\hat{\theta}$ of the linear target mapping, we can constrain the parameter $\hat{\theta}$ in $\Lambda$ and approximate it to the optimal parameter $\theta^*$ during the optimization process by:

$$Min \quad \mathcal{L}(z^i \cdot \hat{\theta}, y^i) - \sum_{m=1}^{M} w^m \mathcal{L}(z_m^i \cdot \theta_m, y^i). \tag{6}$$

Nevertheless, IID is established based on the IF framework, which renders the unavailability of LF-based Cross-Entropy loss function (i.e., $\mathcal{L}(z_m^i \cdot \theta_m, y^i)$), ultimately leading to incalculable Eq.(6). But we note that recent research regarding information theory Tishby et al. (2000); Federici et al. (2020); Li et al. (2024) maximizes mutual information by minimizing Cross-Entropy function, which manifests that higher mutual information $I(z^i; y^i)$ indicates lower $\mathcal{L}(z^i \cdot \hat{\theta}, y^i)$. Drawing inspiration from this finding, we inversely implement Eq.(6) by

$$Max \quad I(z^i; y^i) - \sum_{m=1}^{M} I(z_m^i; y^i), \tag{7}$$

where $I(z^i; y^i) = \int \int p(z^i, y^i) \log \left[ \frac{p(y^i|z^i)}{p(y^i)} \right] dz^i dy^i$. Although Eq.(7) is a necessary but not sufficient condition of Eq.(6), achieving the optimization objective through the necessary condition is

practical and general Jiang & Veitch (2022); Zhang et al. (2024), and the empirical results in **Section 5** confirm the effectiveness of such an implementation.

Compared to Eq.(6), we omit the multimodal fusion weight $w^m$ in Eq.(7) since maximizing $-I(\boldsymbol{z}_m; y)$ equals maximizing $-w^m I(\boldsymbol{z}_m; y)$ with $w^m > 0$. Eventually, we can implement Eq.(7) by minimizing the loss function:

$$\mathcal{L}_{ic} = \sum_{i=1}^{N} \left( -\log q_{\boldsymbol{\theta}}(y^i|\boldsymbol{z}^i) + \lambda KL\left(\mathcal{N}_{\boldsymbol{z}^i}||\mathcal{N}\right) - \sum_{m=1}^{M} \left[ \log q_{\boldsymbol{\theta}}(y^i|\boldsymbol{z}_m^i) - \lambda KL\left(\mathcal{N}_{\boldsymbol{z}_m^i}||\mathcal{N}\right) \right] \right).$$
(8)

The derivation of $\mathcal{L}_{ic}$ is detailed in **Appendix** A.2.4. $q_{\boldsymbol{\theta}}(\cdot|\cdot)$ is the variational approximation of $p(\cdot|\cdot)$, which is calculated by the target mapping. $\lambda$ is a trade-off hyper-parameter. $KL(\cdot)$ is Kullback-Leibler divergence Van Erven et al. (2014). $\mathcal{N}_{\boldsymbol{z}^i}(\mathcal{N}_{\boldsymbol{z}_m^i})$ is a Gaussian distribution fitted by the mean and variance of $\boldsymbol{z}^i(\boldsymbol{z}_m^i)$. $\mathcal{N}$ is the standard Gaussian distribution.

## 4.2 DISTRIBUTION COHERING WITH RESTRICTED ISOMETRIC DIMENSIONALITY REDUCTION

A direct approach to minimize $\sum_{m=1}^{M} \mathbb{E}(w^m) \mathcal{D}_{\mathcal{M}}(\mu_m, \mu)$ is obtaining the distribution barycenter Agueh & Carlier (2011), but such a strategy is very computationally expensive Nguyen et al. (2025). According to our derivation in **Appendix** A.2.5, utilizing the IF approach specified in Eq.(5) for the integration of unimodal features, the following equation holds for almost all the multimodal scenarios:

$$\sum_{m=1}^{M} \mathbb{E}(w^m) \mathcal{D}_{\mathcal{M}}(\mu_m, \mu) \leq \sum_{m_1, m_2} \mathcal{D}_{\mathcal{M}}(\mu_{m_1}, \mu_{m_2}),$$
(9)

where $m_1, m_2 \in [1, M]$ and $m_1 \neq m_2$. Eq.(9) indicates that the distribution incoherence term is bounded by the inter-modality distribution discrepancy, thus we can achieve distribution cohering by minimizing the right-hand side of Eq.(9). Considering that Wasserstein distance possesses the requisite properties of complete distribution distance metric, we determine to accomplish $\mathcal{D}_{\mathcal{M}}$ by Wasserstein distance (detailed in **Appendix** A.3). Sinkhorn algorithm Cuturi (2013) can achieve precise Wasserstein distance calculation. But in practice, performing Sinkhorn algorithm on high-dimensional features is problematic for its excessive computational complexity.

We ascertain the underlying cause by first restating the operating mechanism of canonical Sinkhorn algorithm. Given two probability distribution $p_1, p_2$ with discrete supports $\boldsymbol{u} = \{u_j\}_{j=1}^{n_1}, \boldsymbol{v} = \{v_k\}_{k=1}^{n_2} (\sum_{j=1}^{n_1} u_j = 1$ and $\sum_{k=1}^{n_2} v_k = 1)$, Wasserstein distance can be calculated as follows:

$$\mathcal{W}(p_1, p_2) = min \sum_{j=1}^{n_1} \sum_{k=1}^{n_2} T_{jk} C_{jk}, \text{ subject to } T \in \mathbb{R}_+^{n_1 \times n_2}, T\mathbf{1}_{n_2} = \boldsymbol{u}, T^\top \mathbf{1}_{n_1} = \boldsymbol{v}.$$
(10)

$T$ is the transport plan and $C_{jk}$ evaluates the distance between $u_j$ and $u_k$. During the iterative computation of the optimal transport plan $T$, each element of the matrix $C$ is derived from the pairwise distance between features. Consequently, a large feature dimensionality incurs a substantial computational complexity, which renders the Sinkhorn algorithm computationally problematic for high-dimensional features.

Inspired by the studies Wright & Ma (2022); Radhakrishnan et al. (2025) indicating that the features of data from multiple sources (such as signal, image, and so on) are generally sparse in the frequency domain, we opt to transform high-dimensional sparse features into low-dimensional dense features to accelerate Sinkhorn algorithm. Beyond improving computational efficiency, to mitigate the degradation of Wasserstein distance estimation precision caused by dimensionality reduction, we particularly impose a dimensionality reduction matrix with Restricted Isometry Property (RIP)[2], and can maintain the geometric structure of features during the dimensionality reduction.

Specifically, we first employ fast Fourier transform to transform feature $\boldsymbol{z}_m^i$ into the frequency domain: $\hat{\boldsymbol{z}}_m^i = \mathcal{F}(\boldsymbol{z}_m^i) = \int_{\mathbb{R}^d} f(\boldsymbol{t}) e^{-2\pi i \cdot \boldsymbol{z}_m^i} d\boldsymbol{t}$, where $f(\boldsymbol{t})$ is the Fourier series expansion of $\boldsymbol{z}_m^i$. We further strengthen the sparsity of $\hat{\boldsymbol{z}}_m^i$ by preserve the top-$d_1$ ($d_1 \ll d$) principal components by

$$\hat{z}_{m,n}^i = \begin{cases} \hat{z}_{m,n}^i, & \text{if } |\hat{z}_{m,n}^i| \geq \tau \\ 0, & \text{if } |\hat{z}_{m,n}^i| < \tau \end{cases}.$$
(11)

---

[2] A transform $\boldsymbol{A}$ satisfies RIP if $(1 - \delta') \|\boldsymbol{x}\|_2^2 \leq \|\boldsymbol{A}x\|_2^2 \leq (1 + \delta') \|\boldsymbol{x}\|_2^2$.

Table 1: Results on four vision-language datasets. **Bold** represents the best results. D stands for dynamic fusion, i.e., the fusion weight $w^m$ is a function of $x^m$. In contrast, the $w^m$ is a constant in the static fusion (S) method. We obtain the $p$-value of IID-P by performing the student $t$-test between IID-P and PDF, the same applies to the $p$-value of IID-Q and IID-L.

| Baseline | Type | MVSA-Single | | MVSA-Multiple | | HFM | | Food101 | |
|---|---|---|---|---|---|---|---|---|---|
| | | Avg | Worst | Avg | Worst | Avg | Worst | Avg | Worst |
| Bow | S | $48.79 \pm 7.05$ | 35.45 | $64.78 \pm 0.81$ | 64.18 | $74.22 \pm 0.87$ | 73.25 | $82.50 \pm 0.18$ | 82.32 |
| Img | S | $64.12 \pm 1.23$ | 62.04 | $67.04 \pm 0.49$ | 66.65 | $74.74 \pm 0.38$ | 74.36 | $64.62 \pm 0.40$ | 64.22 |
| BERT | S | $75.61 \pm 0.53$ | 74.76 | $69.39 \pm 0.37$ | 69.18 | $85.34 \pm 0.46$ | 84.86 | $86.46 \pm 0.05$ | 86.42 |
| Late-fusion | S | $76.88 \pm 1.30$ | 74.76 | $67.94 \pm 0.56$ | 67.41 | $85.51 \pm 0.18$ | 85.31 | $90.69 \pm 0.12$ | 90.58 |
| C-Bow | S | $64.08 \pm 1.54$ | 62.04 | $67.35 \pm 0.20$ | 67.24 | $76.53 \pm 0.23$ | 76.28 | $70.77 \pm 0.09$ | 70.68 |
| C-BERT | S | $65.59 \pm 1.33$ | 64.74 | $67.71 \pm 1.06$ | 66.59 | $85.82 \pm 1.06$ | 84.76 | $88.20 \pm 0.34$ | 87.81 |
| MMBT | D | $78.50 \pm 0.40$ | 78.04 | $69.88 \pm 0.31$ | 69.71 | $85.39 \pm 0.34$ | 85.01 | $91.52 \pm 0.10$ | 91.38 |
| TMC | D | $74.87 \pm 2.24$ | 71.10 | $68.41 \pm 0.16$ | 68.29 | $85.18 \pm 0.79$ | 84.55 | $89.86 \pm 0.07$ | 89.80 |
| DYNMM | D | $79.07 \pm 0.53$ | 78.23 | $68.55 \pm 0.20$ | 68.32 | $85.32 \pm 0.42$ | 84.96 | $92.59 \pm 0.07$ | 92.50 |
| LCKD | S | $62.44 \pm 0.30$ | 62.27 | $66.02 \pm 0.13$ | 65.93 | $82.43 \pm 0.53$ | 81.87 | $85.32 \pm 0.36$ | 84.26 |
| QMF | D | $78.07 \pm 1.10$ | 76.30 | $68.67 \pm 0.27$ | 68.41 | $85.87 \pm 0.23$ | 85.66 | $92.92 \pm 0.11$ | 92.72 |
| UniCODE | S | $66.97 \pm 0.39$ | 65.94 | $66.21 \pm 0.32$ | 65.98 | $83.37 \pm 0.52$ | 82.83 | $88.39 \pm 0.36$ | 87.21 |
| SimMMDG | S | $67.08 \pm 0.35$ | 66.35 | $66.44 \pm 0.23$ | 66.19 | $84.13 \pm 0.41$ | 83.85 | $89.57 \pm 0.38$ | 88.43 |
| PDF | D | $79.94 \pm 0.95$ | 78.42 | $69.54 \pm 0.25$ | 69.26 | $86.03 \pm 0.31$ | 85.77 | $93.32 \pm 0.22$ | 92.84 |
| IID-L | S | $77.78 \pm 1.09$ | 75.89 | $69.32 \pm 0.50$ | 67.84 | $85.94 \pm 0.42$ | 85.41 | $91.93 \pm 0.25$ | 91.21 |
| $p$-value | - | $5.47e^{-3}$ | - | $6.67e^{-3}$ | - | $4.34e^{-2}$ | - | $9.87e^{-3}$ | - |
| IID-Q | D | $80.02 \pm 0.40$ | 79.58 | $71.08 \pm 0.30$ | 70.76 | $86.61 \pm 0.23$ | **86.37** | $93.10 \pm 0.03$ | 93.06 |
| $p$-value | - | $1.07e^{-3}$ | - | $4.74e^{-4}$ | - | $4.97e^{-3}$ | - | $3.69e^{-2}$ | - |
| IID-P | D | $\mathbf{81.13 \pm 0.84}$ | **79.98** | $\mathbf{71.23 \pm 0.44}$ | **70.81** | $\mathbf{86.88 \pm 0.39}$ | 86.32 | $\mathbf{93.73 \pm 0.14}$ | **93.52** |
| $p$-value | - | $3.34e^{-4}$ | - | $9.34e^{-4}$ | - | $6.72e^{-3}$ | - | $1.51e^{-2}$ | - |

$\tau$ is set to the magnitude of the $d_1$-th largest component of $\hat{z}_m^i$, $d_1$ is a hyperparameter, and $n \in [1, d]$ is the dimension index. The enhancement of sparsity not only mitigates the interference of noisy semantics but also alleviates the risk of mapping two disparate high-dimensional features to an identical low-dimensional representation during dimensionality reduction.

Then we design a dimensionality reduction matrix with RIP. Let $\Phi$ denote the Gaussian Random matrix, as the elements sampled from $\mathcal{N}$ are highly uncorrelated with the bases of the Fourier transform, $\Psi = \Phi \mathcal{F}^{-1}$ can be treated as the RIP-preserved dimensionality reduction matrix Wright & Ma (2022). Thus we employ a modality-specific $\Psi_m$ for the dimensionality reduction: $\tilde{z}_m^i = \Psi_m \hat{z}_m^i$, where $\tilde{z}_m^i \in \mathbb{R}^{d_1}$, $\hat{z}_m^i \in \mathbb{C}^d$, and $\Psi_m \in \mathbb{C}^{d_1 \times d}$. Since $d_1 \ll d$ and the RIP of $\Psi_m$, the upper bound of distribution incoherence can be accurately measured with limited computation complexity.

Additionally, the loss of semantics derived from dimensionality reduction is inevitable. To enhance robustness to such an undesirable disturbance, it is plausible to relax the original hard marginal matching constraints, which allows for a flexible assignment of matching mass. Overall, we introduce a relaxed constraint in the form of the Lagrange multiplier method, and ultimately calculate Wasserstein distance between modalities $m_1$ and $m_2$ in the following manner:

$$\widetilde{\mathcal{W}}(\mu_{m_1}, \mu_{m_2}) = \underset{T}{argmin} \sum_{j=1}^{n_1} \sum_{k=1}^{n_2} T_{jk} C_{jk} + \lambda_1 \left[ KL(T\mathbf{1}_{n_2} \| \boldsymbol{u}) + KL(T^\top \mathbf{1}_{n_1} \| \boldsymbol{v}) \right], \quad (12)$$

where $C_{jk} = \| \Psi_m(\tilde{z}_m^j) - \Psi_m(\tilde{z}_m^k) \|_2$. Then distribution incoherence is bounded by $\mathcal{L}_{dc} = \sum_{m_1, m_2} \widetilde{\mathcal{W}}(\mu_{m_1}, \mu_{m_2})$, and the final loss function of IID can be formalized as:

$$\mathcal{L}_{IID} = \alpha \mathcal{L}_{ic} + \beta \mathcal{L}_{dc} + \sum_{i=1}^N \mathcal{L}\left[ f_{\text{IF}}(\boldsymbol{x}^i), y^i \right], \quad (13)$$

$\alpha, \beta$ are the hyperparameters to control the influence of $\mathcal{L}_{ic}$ and $\mathcal{L}_{dc}$. The overall training pipeline is depicted in **Algorithm** 1.

## 5 RESULTS

In this section, we evaluate the performance of IID on three multimodal tasks (i.e., vision-language classification, link prediction, and scene recognition) involving eight datasets. Based on Equation (4), the calculation of $w^m$ affects the predictive capability of MML models. For comprehensive and

Table 2: The link prediction results on two multimodal knowledge graph datasets.

| Model | Type | FB-IMG | | | | WN9-IMG | | | |
| --- | --- | --- | --- | --- | --- | --- | --- | --- | --- |
| | | MRR | H@1 | H@3 | H@10 | MRR | H@1 | H@3 | H@10 |
| TransE | Unimodal | 0.712 | 0.618 | 0.781 | 0.859 | 0.865 | 0.765 | 0.816 | 0.871 |
| DistMult | | 0.706 | 0.606 | 0.742 | 0.808 | 0.901 | 0.895 | 0.913 | 0.925 |
| ComplEx | | 0.808 | 0.757 | 0.845 | 0.892 | 0.908 | 0.903 | 0.907 | 0.928 |
| RotatE | | 0.794 | 0.744 | 0.827 | 0.883 | 0.910 | 0.901 | 0.915 | 0.926 |
| TransAE | Multimodal | 0.742 | 0.691 | 0.785 | 0.844 | 0.898 | 0.894 | 0.908 | 0.922 |
| IKLR | | 0.755 | 0.698 | 0.794 | 0.857 | 0.901 | 0.900 | 0.912 | 0.928 |
| TBKGE | | 0.812 | 0.764 | 0.850 | 0.902 | 0.912 | 0.904 | 0.914 | 0.931 |
| MMKRL | | 0.827 | 0.783 | 0.857 | 0.906 | 0.913 | 0.905 | 0.917 | 0.932 |
| OTKGE | | 0.843 | 0.799 | 0.876 | 0.916 | 0.923 | 0.911 | 0.930 | 0.947 |
| MMKRL+IID | Multimodal | 0.844 | 0.801 | 0.876 | 0.917 | 0.920 | 0.911 | 0.925 | 0.945 |
| OTKGE+IID | | **0.855** | **0.813** | **0.887** | **0.925** | **0.932** | **0.917** | **0.938** | **0.957** |

fair comparisons, we implement one static IID (i.e., IID-L, the $w^m$ in IID-L is identical to vanilla Late-fusion), and two dynamic IIDs (i.e., IID-P and IID-Q, the $w^m$ in IID-P and IID-Q are identical to PDF and QMF, respectively) across six datasets involved in vision-language classification and scene recognition tasks. As for the link prediction task on two multimodal knowledge graph datasets, we integrate the two proposed modules into competitive IF-based benchmarks. Each experiment is repeated three times. Due to the limited space, datasets, baselines, implementation details, and extended experiments are depicted in **Appendix** A.5 and A.6.

**Quantitative results.** The quantitative results of vision-language classification and scene recognition are depicted in Tables 1 and 3, respectively. All comparisons are performed in terms of both the average and worst-case accuracy metrics. Under these metrics, the proposed IID-Q and IID-P attain the Top-2 performance on all six datasets. This outcome underscores the superior generalization capability of our models in comparison to the chosen benchmarks. Additionally, PDF and IID-P (QMF and IID-Q, Late-fusion and IID-L) adopt the identical implementation of fusion weights, thus the comparisons between these pairs can further verify the effectiveness of the proposed two modules. According to the quantitative results, the proposed IF-based models consistently outperform

Table 3: Results of scene recognition.

| Baseline | Type | NYU Depth V2 | | SUN RGB-D | |
| --- | --- | --- | --- | --- | --- |
| | | Avg | Worst | Avg | Worst |
| RGB | S | $62.65 \pm 1.22$ | 62.54 | $52.99 \pm 0.88$ | 56.51 |
| Depth | S | $63.30 \pm 0.48$ | 61.01 | $56.78 \pm 0.19$ | 51.32 |
| Late-fusion | S | $69.14 \pm 0.67$ | 68.35 | $62.00 \pm 0.15$ | 60.55 |
| Concat | S | $70.31 \pm 0.80$ | 69.42 | $62.48 \pm 0.50$ | 61.19 |
| Align | S | $70.31 \pm 1.28$ | 68.50 | $61.12 \pm 0.61$ | 60.12 |
| MMTM | D | $71.04 \pm 0.41$ | 70.18 | $61.72 \pm 0.67$ | 60.94 |
| TMC | D | $71.06 \pm 0.76$ | 69.57 | $60.68 \pm 0.24$ | 60.31 |
| LCKD | S | $68.01 \pm 0.31$ | 66.15 | $56.43 \pm 0.56$ | 56.32 |
| QMF | D | $70.09 \pm 0.97$ | 68.81 | $62.09 \pm 0.56$ | 61.30 |
| UniCODE | S | $70.12 \pm 0.37$ | 68.74 | $59.21 \pm 0.55$ | 58.55 |
| SimMMDG | S | $71.34 \pm 0.32$ | 70.29 | $60.54 \pm 0.50$ | 60.31 |
| PDF | D | $71.37 \pm 0.76$ | 70.18 | $62.34 \pm 0.43$ | 61.88 |
| IID-L | S | $69.87 \pm 0.78$ | 68.78 | $62.31 \pm 0.21$ | 60.76 |
| $p$-value | - | $9.12e^{-3}$ | - | $4.84e^{-2}$ | - |
| IID-Q | D | $71.61 \pm 0.50$ | 71.25 | $62.92 \pm 0.13$ | **62.78** |
| $p$-value | - | $5.84e^{-3}$ | - | $3.81e^{-4}$ | - |
| IID-P | D | **$72.04 \pm 0.55$** | **71.49** | $62.99 \pm 0.24$ | 62.71 |
| $p$-value | - | $1.89e^{-3}$ | - | $8.93e^{-3}$ | - |

their LF-based counterparts. Furthermore, we conduct the Student $t$-test Kim (2015), in which $p < 0.05$ indicates a significant difference between the two groups of accuracy samples. Based on the results of Student $t$-test in Tables 1 and 3, the $p$-values are all less than $0.05$, thus we can attribute the performance improvement to the two proposed techniques, rather than the randomness.

We employ four evaluation metrics to assess the performance on the link prediction task: the Mean Reciprocal Rank (MRR) of the correct entities, and Hits@$k$, defined as the proportion of test instances in which the correct entity is ranked within the top-$k$ predictions, where $k \in \{1, 3, 10\}$. Big MRR and Hits@$k$ indicate a good result. We present the results of the link prediction task in Table 2. For the two existing IF-based benchmarks, MMKRL Lu et al. (2022) and OTKGE Cao et al. (2022), we observe that integrating IID further improves their performance. In particular, OTKGE + IID achieves state-of-the-art results on both multimodal knowledge graph datasets. The results indicate that the proposed method can serve as a plug-and-play module to enhance the performance of approaches based on IF framework. Overall, the quantitative results on eight datasets, covering three distinct tasks with diverse modality combinations, validate the effectiveness of the proposed method and further attest to its generalization capability.

**Ablation study**. To investigate the contribution of each ingredient, two variants are trained for justification: i) w/o D removes the distribution cohering with restricted isometric dimensionality reduction module; ii) w/o I excludes the informatic constraint on the linear target mapping. The results of the ablation study are depicted in Table 4. It can be seen that the performance of IID drops

Table 4: The ablation study on six benchmark datasets.

| Dataset | w/o D | w/o I | IID-L | w/o D | w/o I | IID-Q | w/o D | w/o I | IID-P |
|---|---|---|---|---|---|---|---|---|---|
| MVSA-Single | $77.42 \pm 0.73$ | $77.47 \pm 1.10$ | $\mathbf{77.78 \pm 1.09}$ | $79.32 \pm 0.73$ | $78.93 \pm 0.97$ | $\mathbf{80.02 \pm 0.40}$ | $80.79 \pm 0.80$ | $80.46 \pm 0.73$ | $\mathbf{81.13 \pm 0.84}$ |
| MVSA-Multiple | $69.03 \pm 0.41$ | $69.11 \pm 0.76$ | $\mathbf{69.32 \pm 0.50}$ | $69.59 \pm 1.20$ | $70.67 \pm 0.29$ | $\mathbf{71.08 \pm 0.30}$ | $70.13 \pm 0.52$ | $70.61 \pm 0.45$ | $\mathbf{71.23 \pm 0.44}$ |
| HFM | $85.77 \pm 0.38$ | $85.71 \pm 0.53$ | $\mathbf{85.94 \pm 0.42}$ | $86.54 \pm 0.25$ | $86.22 \pm 0.10$ | $\mathbf{86.61 \pm 0.23}$ | $86.61 \pm 0.37$ | $86.35 \pm 0.59$ | $\mathbf{86.88 \pm 0.39}$ |
| Food101 | $91.43 \pm 0.11$ | $91.35 \pm 0.34$ | $\mathbf{91.93 \pm 0.25}$ | $92.98 \pm 0.04$ | $93.01 \pm 0.03$ | $\mathbf{93.10 \pm 0.03}$ | $93.58 \pm 0.07$ | $93.60 \pm 0.15$ | $\mathbf{93.73 \pm 0.14}$ |
| NYU Depth V2 | $69.39 \pm 0.70$ | $69.41 \pm 0.91$ | $\mathbf{69.87 \pm 0.78}$ | $70.95 \pm 0.40$ | $70.48 \pm 0.97$ | $\mathbf{71.61 \pm 0.50}$ | $71.75 \pm 0.48$ | $71.58 \pm 0.92$ | $\mathbf{72.04 \pm 0.55}$ |
| SUN RGB-D | $62.18 \pm 0.17$ | $62.25 \pm 0.37$ | $\mathbf{62.31 \pm 0.24}$ | $62.68 \pm 0.07$ | $62.65 \pm 0.31$ | $\mathbf{62.92 \pm 0.13}$ | $62.77 \pm 0.19$ | $62.53 \pm 0.38$ | $\mathbf{62.99 \pm 0.24}$ |

regardless of which module is removed, suggesting that each proposed technique has a significant impact on the predictive capability of IID.

**Empirical demonstrations of the theoretical derivations.** The design of IID is grounded in two theoretical derivations: $(i)$ $\mathcal{L}_{ic}$ can restrict the parameter of linear target mapping in $\Lambda$ and render the parameter to approximate the optimal parameter $\boldsymbol{\theta}^*$ during the optimization process; $(ii)$ $\mathcal{L}_{dc}$ reduces the generalization error of IID by mitigating the distribution incoherence,

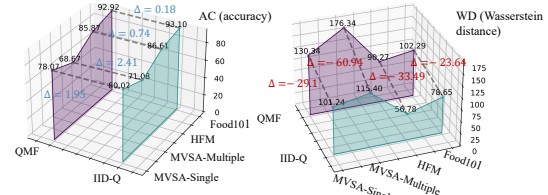

Figure 4: The empirical demonstrations of theoretical derivations (2/2).

thus enhancing the classification performance. Then, we substantiate the correctness of our theoretical derivations with the experimental results. In Figure 5, with informatic constraint $\mathcal{L}_{ic}$ on the linear target mapping, the performance improvement of the IF-based methods compared to the LF-based methods increases, which indicates that $\mathcal{L}_{ic}$ can lead the initial parameter of the linear target mapping to approach the theoretically optimal $\boldsymbol{\theta}^*$. In Figure 4, we present the test classification accuracy of IID-Q and QMF (the left subfigure of Figure 4), along with the mean Wasserstein distance between various unimodal features for each batch of samples (the right subfigure of Figure 4). The results confirm that the classification performance improves as Wasserstein distance decreases. This demonstrates the validity of our theoretical derivation, specifically that eliminating the distribution incoherence contributes to enhanced model prediction performance on unknown test sets.

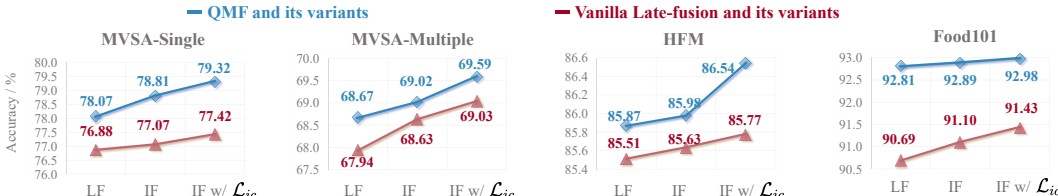

Figure 5: The empirical demonstrations of the theoretical derivations (1/2). In this figure, LF denotes the LF-based models (e.g., QMF and Late-fusion), IF denotes the LF framework is replaced by the IF framework, and IF w/ $\mathcal{L}_{ic}$ means imposing the informatic constraint on the linear target mapping.

## 6 CONCLUSION

In this paper, we rethink the prevalent IF and LF paradigms in MML from a fine-grained dimensional perspective. The complete theoretical derivations sufficiently establish the superiority of IF over LF under a specific constraint. Based on the general $K$-Lipschitz continuity assumption on the linear target mapping, we formalize the generalization error upper bound of IF-based methods, which indicates that the generalization error upper bound can be further decreased by mitigating the distribution incoherence. Motivated by these theoretical insights, we propose IID, an IF-based approach which incorporates linear target mapping with informatic constraint and distribution cohering with restricted isometric dimensionality reduction. Empirical evidence proves that our findings are solid and IID is generally effective.

## ACKNOWLEDGMENTS

This work is supported by National Natural Science Foundation of China No. 62406313, Postdoctoral Fellowship Program of China Postdoctoral Science Foundation, Grant No. YJB20250283.

## REPRODUCIBILITY STATEMENT

We ensure reproducibility by detailing experimental settings, datasets, and hyperparameters in both **Section** 5 and **Appendix** A.5. The source code is included in the supplementary materials to reproduce the proposed method.

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

# A APPENDIX

## A.1 USE OF LLMS

For this manuscript, large language models are utilized exclusively for linguistic polishing. Beyond this function, large language models make no substantive contributions to the conception, analysis, or completion of this work.

## A.2 THEORETICAL DERIVATION

### A.2.1 PROOF OF THEOREM 1

In this subsection, we demonstrate that a vanilla linear target mapping can establish the superiority of IF over LF. For the binary classification task, the activation function is Sigmoid function:

$$\sigma(x) = \frac{1}{1 + e^{-x}}, \tag{14}$$

and the predicted label is $\hat{y} = \begin{cases} 1, & \text{if the prediction logits} > 0 \\ 0, & \text{else} \end{cases}$. We have

$$
\begin{aligned}
\frac{\partial \mathcal{L}(f(\boldsymbol{x}), y)}{\partial f(\boldsymbol{x})} &= \frac{\partial \mathcal{L}(f(\boldsymbol{x}), y)}{\partial \sigma[f(\boldsymbol{x})]} \cdot \frac{\partial \sigma[f(\boldsymbol{x})]}{\partial f(\boldsymbol{x})} \\
&= \frac{\partial\{-y \ln \sigma[f(\boldsymbol{x})] - (1 - y) \ln\{1 - \sigma[f(\boldsymbol{x})]\}\}}{\partial \sigma[f(\boldsymbol{x})]} \cdot \frac{\partial \sigma[f(\boldsymbol{x})]}{\partial f(\boldsymbol{x})} \\
&= \left\{ -y \frac{1}{\sigma[f(\boldsymbol{x})]} + (1 - y) \frac{1}{\{1 - \sigma[f(\boldsymbol{x})]\}} \right\} \cdot \sigma[f(\boldsymbol{x})]\{1 - \sigma[f(\boldsymbol{x})]\} \\
&= \frac{\sigma[f(\boldsymbol{x})] - y}{\sigma[f(\boldsymbol{x})]\{1 - \sigma[f(\boldsymbol{x})]\}} \cdot \sigma[f(\boldsymbol{x})]\{1 - \sigma[f(\boldsymbol{x})]\} = \sigma[f(\boldsymbol{x})] - y,
\end{aligned}
\tag{15}
$$

and $\sigma[f(\boldsymbol{x})] \in (0, 1)$. Therefore, the loss function $\mathcal{L}(\cdot, \cdot)$ is a monotonically decreasing function for the samples with the label $y = 1$, and an increasing function for samples with the label $y = 0$.

As mentioned in **Section** 3, the logits of LF can be formalized as

$$
\begin{aligned}
f_{\mathrm{LF}}(\boldsymbol{x}) &= w^1(\boldsymbol{z}_1 \cdot \boldsymbol{\theta}_1) + w^2(\boldsymbol{z}_2 \cdot \boldsymbol{\theta}_2) \\
&= w^1 \boldsymbol{z}_{1,S_1} \cdot \boldsymbol{\theta}_{1,S_1} + w^1 \boldsymbol{z}_{1,N_1} \cdot \boldsymbol{\theta}_{1,N_1} + w^2 \boldsymbol{z}_{2,S_2} \cdot \boldsymbol{\theta}_{2,S_2} + w^2 \boldsymbol{z}_{2,N_2} \cdot \boldsymbol{\theta}_{2,N_2},
\end{aligned}
\tag{16}
$$

which equals to

$$
f_{\mathrm{LF}}(\boldsymbol{x}) = w^1 \sum_{i \in S_1} z_{1,i} \theta_{1,i} + w^1 \sum_{j \in N_1} z_{1,j} \theta_{1,j} + w^2 \sum_{k \in S_2} z_{2,k} \theta_{2,k} + w^2 \sum_{h \in N_2} z_{2,h} \theta_{2,h}. \tag{17}
$$

The prediction logits of IF can be formalized as

$$
\begin{aligned}
f_{\mathrm{IF}}(\boldsymbol{x}) &= \boldsymbol{z} \cdot \boldsymbol{\theta} \\
&= \boldsymbol{z}_{\mathbb{D}_{S_1 S_2}} \cdot \boldsymbol{\theta}_{\mathbb{D}_{S_1 S_2}} + \boldsymbol{z}_{\mathbb{D}_{S_1 N_2}} \cdot \boldsymbol{\theta}_{\mathbb{D}_{S_1 N_2}} + \boldsymbol{z}_{\mathbb{D}_{N_1 S_2}} \cdot \boldsymbol{\theta}_{\mathbb{D}_{N_1 S_2}} + \boldsymbol{z}_{\mathbb{D}_{N_1 N_2}} \cdot \boldsymbol{\theta}_{\mathbb{D}_{N_1 N_2}}.
\end{aligned}
\tag{18}
$$

Analogously, Eq.(18) can be rewritten as

$$
\begin{aligned}
f_{\text{IF}}(\boldsymbol{x}) &= \sum_{i' \in \mathbb{D}_{S_1 S_2}} z_{i'} \theta_{i'} + \sum_{j' \in \mathbb{D}_{S_1 N_2}} z_{j'} \theta_{j'} + \sum_{k' \in \mathbb{D}_{N_1 S_2}} z_{k'} \theta_{k'} + \sum_{h' \in \mathbb{D}_{N_1 N_2}} z_{h'} \theta_{h'} \\
&= \sum_{i' \in \mathbb{D}_{S_1 S_2}} \theta_{i'}(w^1 z_{1,i'} + w^2 z_{2,i'}) + \sum_{j' \in \mathbb{D}_{S_1 N_2}} \theta_{j'}(w^1 z_{1,j'} + w^2 z_{2,j'}) \\
&\quad + \sum_{k' \in \mathbb{D}_{N_1 S_2}} \theta_{k'}(w^1 z_{1,k'} + w^2 z_{2,k'}) + \sum_{h' \in \mathbb{D}_{N_1 N_2}} \theta_{h'}(w^1 z_{1,h'} + w^2 z_{2,h'}) \\
&= w^1 \Big( \sum_{i' \in \mathbb{D}_{S_1 S_2}} \theta_{i'} z_{1,i'} + \sum_{j' \in \mathbb{D}_{S_1 N_2}} \theta_{j'} z_{1,j'} \Big) + w^1 \Big( \sum_{k' \in \mathbb{D}_{N_1 S_2}} \theta_{k'} z_{1,k'} + \sum_{h' \in \mathbb{D}_{N_1 N_2}} \theta_{h'} z_{1,h'} \Big) \\
&\quad + w^2 \Big( \sum_{i' \in \mathbb{D}_{S_1 S_2}} \theta_{i'} z_{2,i'} + \sum_{k' \in \mathbb{D}_{N_1 S_2}} \theta_{k'} z_{2,k'} \Big) + w^2 \Big( \sum_{j' \in \mathbb{D}_{S_1 N_2}} \theta_{j'} z_{2,j'} + \sum_{h' \in \mathbb{D}_{N_1 N_2}} \theta_{h'} z_{2,h'} \Big).
\end{aligned}
\tag{19}
$$

Given the Bayes optimal hypothesis $f^*$, which achieves the infimum of the errors $\mathcal{R}^*$ on $\mathcal{D}$, i.e.:

$$
f^* = \underset{f}{argmin} \ \mathcal{R}(f) = \underset{f}{argmin} \ \mathbb{E}_{(\boldsymbol{x},y) \sim \mathcal{D}} \left[ \mathcal{L}(f(\boldsymbol{x}), y) \right].
\tag{20}
$$

Equation $\mathcal{L}(f_{\text{LF}}(\boldsymbol{x}), y) \geq \mathcal{L}(f^*(\boldsymbol{x}), y)$ holds universally. Let $\Delta, \Delta_1, \Delta_2, \Delta_3$, and $\Delta_4$ be five scalars that are positive correlated with $y - \delta_1$ ($\Delta, \Delta_1, \Delta_2, \Delta_3, \Delta_4 \propto y - \delta_1$), where $\delta > 0$ is an arbitrarily small positive constant and $\Delta = \sum_{i=1}^{4} \Delta_i$.

Obviously, we have the conclusion: for $\forall \epsilon \in [0, \| \mathcal{L}(f^*(\boldsymbol{x}), y) - \mathcal{L}(f_{\text{LF}}(\boldsymbol{x}), y) \|]$, there exists $f_{\text{LF}}(\boldsymbol{x}) + \Delta$ such that the classification error $\mathcal{L}(f_{\text{LF}}(\boldsymbol{x}) + \Delta, y) = \mathcal{L}(f_{\text{LF}}(\boldsymbol{x}), y) - \epsilon$. Thus core challenge lies in proving the existence of $\boldsymbol{\theta}$ which makes $f_{\boldsymbol{\theta},IF}(\boldsymbol{x}) = f_{\text{LF}}(\boldsymbol{x}) + \Delta$.

Considering the following linear equations:

$$
\begin{cases}
\displaystyle \sum_{i' \in \mathbb{D}_{S_1 S_2}} \theta_{i'} z_{1,i'} + \sum_{j' \in \mathbb{D}_{S_1 N_2}} \theta_{j'} z_{1,j'} = \sum_{i \in S_1} z_{1,i} \theta_{1,i} + \Delta_1 \\[3mm]
\displaystyle \sum_{k' \in \mathbb{D}_{N_1 S_2}} \theta_{k'} z + \sum_{h' \in \mathbb{D}_{N_1 N_2}} \theta_{h'} z_{1,h'} = \sum_{j \in N_1} z_{1,j} \theta_{1,j} + \Delta_2 \\[3mm]
\displaystyle \sum_{i' \in \mathbb{D}_{S_1 S_2}} \theta_{i'} z_{2,i'} + \sum_{k' \in \mathbb{D}_{N_1 S_2}} \theta_{k'} z_{2,k'} = \sum_{k \in S_2} z_{2,k} \theta_{2,k} + \Delta_3 \\[3mm]
\displaystyle \sum_{j' \in \mathbb{D}_{S_1 N_2}} \theta_{j'} z_{2,j'} + \sum_{h' \in \mathbb{D}_{N_1 N_2}} \theta_{h'} z_{2,h'} = \sum_{h \in N_2} z_{2,h} \theta_{2,h} + \Delta_4
\end{cases},
\tag{21}
$$

we treat the parameters of linear target mappings in IF as the coefficients to be determined, and we denote the $i$-th element of the set $S$ by $i_S$, then we have:

$$
\boldsymbol{A}
\begin{bmatrix}
\theta_{1_{\mathbb{D}_{S_1 S_2}}} \\
\vdots \\
\theta_{|\mathbb{D}_{S_1 S_2}|_{\mathbb{D}_{S_1 S_2}}} \\
\theta_{1_{\mathbb{D}_{S_1 N_2}}} \\
\vdots \\
\theta_{|\mathbb{D}_{S_1 N_2}|_{\mathbb{D}_{S_1 N_2}}} \\
\theta_{1_{\mathbb{D}_{N_1 S_2}}} \\
\vdots \\
\theta_{|\mathbb{D}_{N_1 S_2}|_{\mathbb{D}_{N_1 S_2}}} \\
\theta_{1_{\mathbb{D}_{N_1 N_2}}} \\
\vdots \\
\theta_{|\mathbb{D}_{N_1 N_2}|_{\mathbb{D}_{N_1 N_2}}}
\end{bmatrix}
\begin{matrix}
\left.\begin{matrix} \\ \\ \\ \end{matrix}\right\} \text{index of } \mathbb{D}_{S_1 S_2} \\
\left.\begin{matrix} \\ \\ \\ \end{matrix}\right\} \text{index of } \mathbb{D}_{S_1 N_2} \\
\left.\begin{matrix} \\ \\ \\ \end{matrix}\right\} \text{index of } \mathbb{D}_{N_1 S_2} \\
\left.\begin{matrix} \\ \\ \\ \end{matrix}\right\} \text{index of } \mathbb{D}_{N_1 N_2}
\end{matrix}
=
\begin{bmatrix}
\sum_{i \in S_1} z_{1,i} \theta_{1,i} + \Delta_1 \\
\sum_{j \in N_1} z_{1,j} \theta_{1,j} + \Delta_2 \\
\sum_{k \in S_2} z_{2,k} \theta_{2,k} + \Delta_3 \\
\sum_{h \in N_2} z_{2,h} \theta_{2,h} + \Delta_4
\end{bmatrix},
\tag{22}
$$

where $A$ equals

$$
\begin{bmatrix}
z_{1,1_{\mathbb{D}_{S_1 S_2}}} \cdots z_{1,|\mathbb{D}_{S_1 S_2}|_{\mathbb{D}_{S_1 S_2}}} & z_{1,1_{\mathbb{D}_{S_1 N_2}}} \cdots z_{1,|\mathbb{D}_{S_1 N_2}|_{\mathbb{D}_{S_1 N_2}}} & \mathbf{0} & \mathbf{0} \\
\mathbf{0} & \mathbf{0} & z_{1,1_{\mathbb{D}_{N_1 S_2}}} \cdots z_{1,|\mathbb{D}_{N_1 S_2}|_{\mathbb{D}_{N_1 S_2}}} & z_{1,1_{\mathbb{D}_{N_1 N_2}}} \cdots z_{1,|\mathbb{D}_{N_1 N_2}|_{\mathbb{D}_{N_1 N_2}}} \\
z_{2,1_{\mathbb{D}_{S_1 S_2}}} \cdots z_{2,|\mathbb{D}_{S_1 S_2}|_{\mathbb{D}_{S_1 S_2}}} & \mathbf{0} & z_{2,\mathbb{D}_{N_1 S_2}} \cdots z_{2,|\mathbb{D}_{N_1 S_2}|_{\mathbb{D}_{N_1 S_2}}} & \mathbf{0} \\
\mathbf{0} & z_{2,1_{\mathbb{D}_{S_1 N_2}}} \cdots z_{2,|\mathbb{D}_{S_1 N_2}|_{\mathbb{D}_{S_1 N_2}}} & \mathbf{0} & z_{2,1_{\mathbb{D}_{N_1 N_2}}} \cdots z_{2,|\mathbb{D}_{N_1 N_2}|_{\mathbb{D}_{N_1 N_2}}}
\end{bmatrix}
$$

and augmented matrix $\tilde{A}$ can be formalized as

$$
\left[
\begin{array}{cccc|c}
z_{1,1_{\mathbb{D}_{S_1 S_2}}} \cdots z_{1,|\mathbb{D}_{S_1 S_2}|_{\mathbb{D}_{S_1 S_2}}} & z_{1,1_{\mathbb{D}_{S_1 N_2}}} \cdots z_{1,|\mathbb{D}_{S_1 N_2}|_{\mathbb{D}_{S_1 N_2}}} & \mathbf{0} & \mathbf{0} & \sum_{i \in S_1} z_{1,i}\theta_{1,i} + \Delta_1 \\
\mathbf{0} & \mathbf{0} & z_{1,1_{\mathbb{D}_{N_1 S_2}}} \cdots z_{1,|\mathbb{D}_{N_1 S_2}|_{\mathbb{D}_{N_1 S_2}}} & z_{1,1_{\mathbb{D}_{N_1 N_2}}} \cdots z_{1,|\mathbb{D}_{N_1 N_2}|_{\mathbb{D}_{N_1 N_2}}} & \sum_{j \in N_1} z_{1,j}\theta_{1,j} + \Delta_2 \\
z_{2,1_{\mathbb{D}_{S_1 S_2}}} \cdots z_{2,|\mathbb{D}_{S_1 S_2}|_{\mathbb{D}_{S_1 S_2}}} & \mathbf{0} & z_{2,\mathbb{D}_{N_1 S_2}} \cdots z_{2,|\mathbb{D}_{N_1 S_2}|_{\mathbb{D}_{N_1 S_2}}} & \mathbf{0} & \sum_{k \in S_2} z_{2,k}\theta_{2,k} + \Delta_3 \\
\mathbf{0} & z_{2,1_{\mathbb{D}_{S_1 N_2}}} \cdots z_{2,|\mathbb{D}_{S_1 N_2}|_{\mathbb{D}_{S_1 N_2}}} & \mathbf{0} & z_{2,1_{\mathbb{D}_{N_1 N_2}}} \cdots z_{2,|\mathbb{D}_{N_1 N_2}|_{\mathbb{D}_{N_1 N_2}}} & \sum_{h \in N_2} z_{2,h}\theta_{2,h} + \Delta_4
\end{array}
\right].
$$

Obviously, the rank of $A$ is equal to the rank of $\tilde{A}$. According to the basic knowledge of Linear Algebra Greub (2012), there must exist a parameter $\theta$ of linear target mapping in IF such that the following equation holds:

$$
A\theta = \begin{bmatrix}
\sum_{i \in S_1} z_{1,i}\theta_{1,i} + \Delta_1 \\
\sum_{j \in N_1} z_{1,j}\theta_{1,j} + \Delta_2 \\
\sum_{k \in S_2} z_{2,k}\theta_{2,k} + \Delta_3 \\
\sum_{h \in N_2} z_{2,h}\theta_{2,h} + \Delta_4
\end{bmatrix}. \tag{23}
$$

Consequently, for $\forall \epsilon \in \left[0, \|\ \mathcal{L}(f^*(\boldsymbol{x}), y) - \mathcal{L}(f_{\mathrm{LF}}(\boldsymbol{x}), y)\ \|\right), y\big)]$, there exists a parameter $\theta$ such that $\mathcal{L}(f_{\theta,IF}(\boldsymbol{x}), y) = \mathcal{L}(f_{\mathrm{LF}}(\boldsymbol{x}) + \Delta, y) = \mathcal{L}(f_{\mathrm{LF}}(\boldsymbol{x}), y) - \epsilon$, which further derives that $\mathcal{L}(f_{\theta,IF}(\boldsymbol{x}), y) < \mathcal{L}(f_{\mathrm{LF}}(\boldsymbol{x}), y)$. We denote the set of parameters satisfying $\mathcal{L}(f_{\theta,IF}(\boldsymbol{x}), y) < \mathcal{L}(f_{\mathrm{LF}}(\boldsymbol{x}), y)$ as $\Lambda$. The proof of Theorem 1 is complete.

### A.2.2 PROOF OF THEOREM 2

Let $(\boldsymbol{x}, y) \sim \mathcal{D}$ denote the multimodal samples, the generalization error is defined as

$$
\mathscr{G} = \mathbb{E}_{(\boldsymbol{x},y) \sim \mathcal{D}}[\mathcal{L}(f(\boldsymbol{x}), y)] = \sum_{i=1}^{|\mathcal{D}|} p(\boldsymbol{x}^i, y^i)\mathcal{L}(f(\boldsymbol{x}^i), y^i), \tag{24}
$$

thus the generalization error of LF and IF can be formalized as:

$$
\mathscr{G}_{LF} = \mathbb{E}_{(\boldsymbol{x},y) \sim \mathcal{D}}[\mathcal{L}(f_{\mathrm{LF}}(\boldsymbol{x}), y)] = \sum_{i=1}^{|\mathcal{D}|} p(\boldsymbol{x}^i, y^i)\mathcal{L}(f_{\mathrm{LF}}(\boldsymbol{x}^i), y^i), \tag{25}
$$

$$
\mathscr{G}_{IF} = \mathbb{E}_{(\boldsymbol{x},y) \sim \mathcal{D}}[\mathcal{L}(f_{\mathrm{IF}}(\boldsymbol{x}), y)] = \sum_{i=1}^{|\mathcal{D}|} p(\boldsymbol{x}^i, y^i)\mathcal{L}(f_{\mathrm{IF}}(\boldsymbol{x}^i), y^i). \tag{26}
$$

Due to $\mathcal{L}(f_{\mathrm{LF}}(\boldsymbol{x}^i), y^i) \geq 0$, $\mathcal{L}(f_{\mathrm{IF}}(\boldsymbol{x}^i), y^i) \geq 0$, and based on the parameter $\theta \in \Lambda$, we have $\mathcal{L}(f_{\mathrm{LF}}(\boldsymbol{x}^i), y^i) \geq \mathcal{L}(f_{\mathrm{IF},\theta}(\boldsymbol{x}^i), y^i)$, therefore the following equation holds:

$$
\sum_{i=1}^{|\mathcal{D}|} p(\boldsymbol{x}^i, y^i)\mathcal{L}(f_{\mathrm{LF}}(\boldsymbol{x}^i), y^i) \geq \sum_{i=1}^{|\mathcal{D}|} p(\boldsymbol{x}^i, y^i)\mathcal{L}(f_{\mathrm{IF},\theta}(\boldsymbol{x}^i), y^i), \tag{27}
$$

which equals to

$$
\mathscr{G}_{IF,\theta} \leq \mathscr{G}_{LF}. \tag{28}
$$

The proof of Theorem 2 has been completed.

### A.2.3 PROOF OF THEOREM 3

In this subsection, based on Assumption 1, we provide the proof of Theorem 3.

Let $z_{m_1}$ and $z_{m_2}$ be the latent features of two arbitrary modalities, which respectively fit the distributions $\mu_{m_1}$ and $\mu_{m_2}$. We have:

$$\hat{\mathbb{E}}[g(z_{m_1})] - \hat{\mathbb{E}}[g(z_{m_2})] = \mathbb{E}_{z_{m_1} \sim \mu_{m_1}}[\mathcal{L}(g(z_{m_1}), y)] - \mathbb{E}_{z_{m_2} \sim \mu_{m_2}}[\mathcal{L}(g(z_{m_2}), y)], \quad (29)$$

According to the Kantorovich-Rubinstein Duality theorem Thickstun (2019); Edwards (2011), we have:

$$\begin{aligned}
&\mathbb{E}_{z_{m_1} \sim \mu_{m_1}}[\mathcal{L}(g(z_{m_1}), y)] - \mathbb{E}_{z_{m_2} \sim \mu_{m_2}}[\mathcal{L}(g(z_{m_2}), y)] \\
&\leq \| \phi \|_{Lip} \, \mathcal{D}_{\mathcal{M}}(\mu_{m_1}, \mu_{m_2}) \\
&\leq K \cdot \mathcal{D}_{\mathcal{M}}(\mu_{m_1}, \mu_{m_2}),
\end{aligned} \quad (30)$$

where $\hat{\mathbb{E}}(\cdot)$ is the emprical error. It's worth noting that if $\mathcal{D}_{\mathcal{M}}$ is not a complete distribution distance metric such as Kullback-Leibler Divergence Kullback (1951), we need to put more discussions on $\hat{\mathbb{E}}[g(z_{m_1})] - \hat{\mathbb{E}}[g(z_{m_2})] \leq K\mathcal{D}_{\mathcal{M}}(\mu_{m_1}, \mu_{m_2})$ because of Kullback-Leibler Divergence's asymmetry i.e., $KL(\mu_{m_1}, \mu_{m_2}) \neq KL(\mu_{m_2}, \mu_{m_1})$.

In Eq.(30), by replacing the feature of $j$-th modality to the fused multimodal feature $z$, we have:

$$\hat{\mathbb{E}}[g(z_{m_1})] - \hat{\mathbb{E}}[g(z)] \leq K\mathcal{D}_{\mathcal{M}}(\mu_{m_1}, \mu_z). \quad (31)$$

$\mu$ is the distribution that multimodal feature $z$ follows, and $i$ can be the index of arbitrary modality, that is, $i \in \{1, 2, \cdots, M\}$, therefore:

$$\hat{\mathbb{E}}[g(z_{m_1})] \leq K\mathcal{D}_{\mathcal{M}}(\mu_{m_1}, \mu) + \hat{\mathbb{E}}(f_{\text{IF}}). \quad (32)$$

Eq.(32) indicates that the empirical error of a certain unimodal modality can be bound by the empirical error of the fused multimodal feature and the distribution distance between the unimodal and the fused feature.

Restating Theorem 1 in Zhang et al. (2023a) and combined with Eq.(32), we have:

$$\begin{aligned}
\mathscr{G}_{IF,\boldsymbol{\theta}} &\leq \sum_{m=1}^{M} \mathbb{E}(w^m)\hat{\mathbb{E}}[g_{\boldsymbol{\theta}}(z_{\boldsymbol{m}})] + \sum_{m=1}^{M} \mathbb{E}(w^m)\mathfrak{R}_m(\mathcal{H}) + \sum_{m=1}^{M} Cov(w^m, \mathcal{L}(g_{\boldsymbol{\theta}}(z_{\boldsymbol{m}}), y)) \\
&\quad + M\sqrt{\frac{ln(1/\delta)}{2N}} \\
&\leq \sum_{m=1}^{M} \mathbb{E}(w^m)[K\mathcal{D}_{\mathcal{M}}(\mu_{z_m}, \mu_z) + \hat{\mathbb{E}}(f_{\text{IF}})] + \sum_{m=1}^{M} \mathbb{E}(w^m)\mathfrak{R}_m(\mathcal{H}) \\
&\quad + \sum_{m=1}^{M} Cov(w^m, \mathcal{L}(g_{\boldsymbol{\theta}}(z_{\boldsymbol{m}}), y)) + M\sqrt{\frac{ln(1/\delta)}{2N}}. \\
&= \sum_{m=1}^{M} [K \cdot \mathbb{E}(w^m)\mathcal{D}_{\mathcal{M}}(\mu_{z_m}, \mu_z) + \text{Error}[w^m, \mathcal{L}(g_{\boldsymbol{\theta}}(z_{\boldsymbol{m}}), y)]] + \hat{\mathbb{E}}(f_{\text{IF}}) + \text{Bias}[\mathfrak{R}(\mathcal{H}), \mathcal{O}(N^{-1/2})].
\end{aligned}$$
$$(33)$$

Thus, the proof of Theorem 3 is complete.

### A.2.4 DERIVATION IN LINEAR TARGET MAPPING WITH INFORMATIC CONSTRAINT

The objective function is:

$$Max \quad I(z; y) - \sum_{m=1}^{M} I(z_m; y). \quad (34)$$

According to Alemi et al. (2017); Tishby et al. (2000); Xiao et al. (2024), to avoid the collapsed representation of $z$ during the learning process of the linear target mapping parameter, we introduce

a regularization term $I(\boldsymbol{x}; \boldsymbol{z})$ and its trade-off coefficient $\lambda$, then we have:

$$
\begin{aligned}
I(\boldsymbol{z}; y) - \sum_{m=1}^{M} I(\boldsymbol{z}_m; y) &= I(\boldsymbol{z}; y) - \lambda I(\boldsymbol{x}; \boldsymbol{z}) + \lambda I(\boldsymbol{x}; \boldsymbol{z}) - \sum_{m=1}^{M} I(\boldsymbol{z}_m; y) \\
&\geq I(\boldsymbol{z}; y) - \lambda I(\boldsymbol{x}; \boldsymbol{z}) + \lambda \sum_{m=1}^{M} I(x_m; \boldsymbol{z}_m) - \sum_{m=1}^{M} I(\boldsymbol{z}_m; y) \\
&= I(\boldsymbol{z}; y) - \lambda I(\boldsymbol{x}; \boldsymbol{z}) - \Big[ \sum_{m=1}^{M} I(\boldsymbol{z}_m; y) - \lambda \sum_{m=1}^{M} I(\boldsymbol{x}; \boldsymbol{z}_m) \Big] \\
&= I(\boldsymbol{z}; y) - \lambda I(\boldsymbol{x}; \boldsymbol{z}) - \sum_{m=1}^{M} \big[ I(\boldsymbol{z}_m; y) - \lambda I(\boldsymbol{x}; \boldsymbol{z}_m) \big].
\end{aligned}
\tag{35}
$$

In the following, we begin examining each term in Eq.(35) from term $I(\boldsymbol{z}; y)$.

$$
I(\boldsymbol{z}; y) = \int \int p(y, \boldsymbol{z}) \log \frac{p(y, \boldsymbol{z})}{p(y) p(\boldsymbol{z})} dy d\boldsymbol{z} = \int \int p(y, \boldsymbol{z}) \log \frac{p(y|\boldsymbol{z})}{p(y)} dy d\boldsymbol{z}.
\tag{36}
$$

Let $q(y|\boldsymbol{z})$ be a variational approximation of $p(y|\boldsymbol{z})$, and we parameterize $q(y|\boldsymbol{z})$ by $\boldsymbol{\theta}$, i.e., $q_{\boldsymbol{\theta}}(y|\boldsymbol{z})$. Based on the fact that the Kullback Leibler (KL) divergence is constantly positive, we have $KL[p(y|z), q_{\boldsymbol{\theta}}(y|\boldsymbol{z})] \geq 0 \Rightarrow \int p(y|\boldsymbol{z}) \log p(y|\boldsymbol{z}) dy \geq \int p(y|\boldsymbol{z}) \log q_{\boldsymbol{\theta}}(y|\boldsymbol{z}) dy$, thus

$$
\begin{aligned}
I(\boldsymbol{z}; y) &\geq \int \int p(y, \boldsymbol{z}) \log \frac{q_{\boldsymbol{\theta}}(y|\boldsymbol{z})}{p(y)} dy d\boldsymbol{z} = \int \int p(y, \boldsymbol{z}) \log q_{\boldsymbol{\theta}}(y|\boldsymbol{z}) dy d\boldsymbol{z} - \int \int p(y, \boldsymbol{z}) \log p(y) dy d\boldsymbol{z} \\
&= \int \int p(y, \boldsymbol{z}) \log q_{\boldsymbol{\theta}}(y|\boldsymbol{z}) dy d\boldsymbol{z} + H(Y) \\
&= \int \int \int p(\boldsymbol{x}) p(y|\boldsymbol{x}) p(\boldsymbol{z}|\boldsymbol{x}) \log q_{\boldsymbol{\theta}}(y|\boldsymbol{z}) dx dy d\boldsymbol{z} + H(Y),
\end{aligned}
\tag{37}
$$

where $H(Y)$ is a constant term and can be ignored. As for the term $I(\boldsymbol{z}; x)$, we have

$$
I(\boldsymbol{z}; \boldsymbol{x}) = \int \int p(\boldsymbol{x}, \boldsymbol{z}) \log \frac{p(\boldsymbol{z}|\boldsymbol{x})}{p(\boldsymbol{z})} d\boldsymbol{z} d\boldsymbol{x}.
\tag{38}
$$

Let $r(\boldsymbol{z})$ be a variational approximation of $p(\boldsymbol{z})$ and we set $r(\boldsymbol{z})$ as the standard Gaussian distribution $\mathcal{N}(0, I)$. Since $KL[p(\boldsymbol{z}), r(\boldsymbol{z})] \geq 0 \implies \int p(\boldsymbol{z}) \log p(\boldsymbol{z}) d\boldsymbol{z} \geq \int p(\boldsymbol{z}) \log r(\boldsymbol{z}) d\boldsymbol{z}$, thus:

$$
I(\boldsymbol{z}; \boldsymbol{x}) \leq \int \int p(\boldsymbol{x}) p(\boldsymbol{z}|\boldsymbol{x}) \log \frac{p(\boldsymbol{z}|\boldsymbol{x})}{r(\boldsymbol{z})} d\boldsymbol{x} d\boldsymbol{z}.
\tag{39}
$$

As a result, we have

$$
\begin{aligned}
&I(\boldsymbol{z}; y) - \lambda I(\boldsymbol{z}; \boldsymbol{x}) \\
&\geq \int \int \int p(\boldsymbol{x}) p(y|\boldsymbol{x}) p(\boldsymbol{z}|\boldsymbol{x}) \log q(y|\boldsymbol{z}) d\boldsymbol{x} dy d\boldsymbol{z} - \lambda \int \int \int p(\boldsymbol{x}) p(\boldsymbol{z}|\boldsymbol{x}) \log \frac{p(\boldsymbol{z}|\boldsymbol{x})}{r(\boldsymbol{z})} d\boldsymbol{x} dy d\boldsymbol{z} \\
&= LB,
\end{aligned}
\tag{40}
$$

$LB$ standards for Lower Bound. Analogously, for the upper bound (UB) of $\sum_{m=1}^{M} [I(\boldsymbol{z}_m; y) - \lambda I(\boldsymbol{x}; \boldsymbol{z}_m)]$, we have

$$
\begin{aligned}
\sum_{m=1}^{M} [I(\boldsymbol{z}_m; y) - \lambda I(\boldsymbol{x}; \boldsymbol{z}_m)] &\leq UB \\
&\leq - \sum_{m=1}^{M} \Big\{ \int \int \int p(\boldsymbol{x}) p(y|\boldsymbol{x}) p(\boldsymbol{z}_m|\boldsymbol{x}) \log q_{\boldsymbol{\theta}}(y|\boldsymbol{z}_m) d\boldsymbol{x} dy d\boldsymbol{z}_m \\
&\quad + \lambda \int \int p(x) p(\boldsymbol{z}_m|x) \log \frac{p(\boldsymbol{z}_m|x)}{r(\boldsymbol{z}_m)} d\boldsymbol{x} d\boldsymbol{z}_m \Big\}.
\end{aligned}
\tag{41}
$$

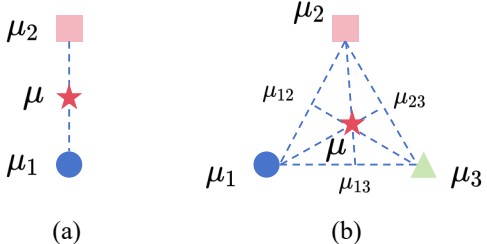

Figure 6: The illustration of the space of probability density functions, in which each point represents a probability distribution.

Then maximizing the $LB - UB$ equals to minimizing the following loss function:

$$\mathcal{L}_{ic} = -\log q_{\boldsymbol{\theta}}(y|\boldsymbol{z}) + \lambda \cdot KL\left(\mathcal{N}_{\boldsymbol{z}}||\mathcal{N}(0,I)\right) - \sum_{m=1}^{M}\left[\log q_{\boldsymbol{\theta}}(y|\boldsymbol{z_m}) + \lambda \cdot KL\left(\mathcal{N}_{\boldsymbol{z_m}}||\mathcal{N}(0,I)\right)\right].$$
(42)

### A.2.5 DERIVATION OF THE UPPER BOUND ON DISTRIBUTION INCOHERENCE

We provide an illustration of the space of probability density functions in Figure 6 to assist our theoretical derivation. Then we demonstrate that the Equation 9 holds for $M = \{2, 3\}$. Let LHS $= \sum_{m=1}^{M} \mathbb{E}(w^m)\mathcal{D}_{\mathcal{M}}(\mu_m, \mu)$ and RHS $= \sum_{m_1, m_2} \mathcal{D}_{\mathcal{M}}(\mu_{m_1}, \mu_{m_2})$.

For $M = 2$, we have

$$\text{LHS} \le \sum_{m=1}^{2} \mathcal{D}_{\mathcal{M}}(\mu_m, \mu) = \mathcal{D}_{\mathcal{M}}(\mu_1, \mu) + \mathcal{D}_{\mathcal{M}}(\mu_2, \mu) = \text{RHS}.$$
(43)

For $M = 3$, we have LHS $\le \sum_{m=1}^{3} \mathcal{D}_{\mathcal{M}}(\mu_m, \mu) \le \mathcal{D}_{\mathcal{M}}(\mu_1, \mu) + \mathcal{D}_{\mathcal{M}}(\mu_2, \mu) + \mathcal{D}_{\mathcal{M}}(\mu_3, \mu)$ and RHS $= \mathcal{D}_{\mathcal{M}}(\mu_1, \mu_2) + \mathcal{D}_{\mathcal{M}}(\mu_1, \mu_3) + \mathcal{D}_{\mathcal{M}}(\mu_2, \mu_3)$. As illustrated in Figure 6, we have

$$\mathcal{D}_{\mathcal{M}}(\mu_1, \mu_2) + \mathcal{D}_{\mathcal{M}}(\mu_2, \mu_{23}) > \mathcal{D}_{\mathcal{M}}(\mu_1, \mu_{23}) = \mathcal{D}_{\mathcal{M}}(\mu_1, \mu) + \mathcal{D}_{\mathcal{M}}(\mu, \mu_{23})$$
(44)

and

$$\mathcal{D}_{\mathcal{M}}(\mu, \mu_{23}) + \mathcal{D}_{\mathcal{M}}(\mu_{23}, \mu_3) > \mathcal{D}_{\mathcal{M}}(\mu, \mu_3).$$
(45)

Then the following equation holds:

$$\mathcal{D}_{\mathcal{M}}(\mu_1, \mu_2) + \mathcal{D}_{\mathcal{M}}(\mu_2, \mu_{23}) + \mathcal{D}_{\mathcal{M}}(\mu, \mu_{23}) + \mathcal{D}_{\mathcal{M}}(\mu_{23}, \mu_3)$$
$$> \mathcal{D}_{\mathcal{M}}(\mu_1, \mu) + \mathcal{D}_{\mathcal{M}}(\mu, \mu_{23}) + \mathcal{D}_{\mathcal{M}}(\mu, \mu_3),$$
(46)

which equals to

$$\mathcal{D}_{\mathcal{M}}(\mu_1, \mu_2) + \mathcal{D}_{\mathcal{M}}(\mu_3, \mu_2) > \mathcal{D}_{\mathcal{M}}(\mu_1, \mu) + \mathcal{D}_{\mathcal{M}}(\mu, \mu_3).$$
(47)

Similarly, we have

$$\mathcal{D}_{\mathcal{M}}(\mu_1, \mu_2) + \mathcal{D}_{\mathcal{M}}(\mu_1, \mu_3) > \mathcal{D}_{\mathcal{M}}(\mu_2, \mu) + \mathcal{D}_{\mathcal{M}}(\mu, \mu_3),$$
$$\mathcal{D}_{\mathcal{M}}(\mu_3, \mu_1) + \mathcal{D}_{\mathcal{M}}(\mu_3, \mu_2) > \mathcal{D}_{\mathcal{M}}(\mu_1, \mu) + \mathcal{D}_{\mathcal{M}}(\mu, \mu_2).$$
(48)

Then we have

$$2 * \text{RHS} > 2 * [\mathcal{D}_{\mathcal{M}}(\mu, \mu_1) + \mathcal{D}_{\mathcal{M}}(\mu, \mu_2) + \mathcal{D}_{\mathcal{M}}(\mu, \mu_3)] > 2 * \text{LHS}.$$
(49)

As a result, Equation 9 holds for $M = \{2, 3\}$, which implies that Equation 9 is applicable for almost all multimodal scenarios according to the recent multimodal learning survey Xu et al. (2023); Yuan et al. (2025) (even the powerful model Bachmann et al. (2024) capable of handling 21 modalities can handle at most 3 modalities at a single time).

---

**Algorithm 1** The training pseudo code of IID.

---

**Input:** The sampled minibatch samples $\{(\boldsymbol{x}^i, y^i) | i \in [1, \cdots, N]\}$ with batchsize $N$ and $\boldsymbol{x}^i = \{x_1^i, x_2^i, \cdots, x_M^i\}$. The latent mappings $h^m(\cdot)$, target mapping $g(\cdot)$ and multimodal fusion weights $w^m$. The hyperparameters $\alpha$ and $\beta$.

**Output:** The loss function of IID, i.e., $\mathcal{L}_{IID}$.

**for** $i=1$ to $N$ **do**

    Obtain unimodal features by $\boldsymbol{z}_m^i = h^m(x_m^i)$ $(m \in [1, M])$;

    Get low-dimensional feature $\tilde{\boldsymbol{z}}_m^i$ of $\boldsymbol{z}_m^i$;

    Calculate the fused multimodal feature $\boldsymbol{z}^i = \sum_{m=1}^{M} w^m \boldsymbol{z}_m^i$;

    Calculate the loss function $\mathcal{L}(f_{\text{IF}}(\boldsymbol{x}^i), y^i)$;

**end**

Calculate the $\mathcal{L}_{ic}$ by Eq.(8);

**for** $m_1, m_2 \in [1, M]$ *and* $m_1 \neq m_2$ **do**

    Get the estimated Wasserstein distance between the features of $m_1$-th modality and $m_2$-th modality by $\widetilde{\mathcal{W}}(\mu_{m_1}, \mu_{m_2})$;

**end**

Calculate the loss function $\mathcal{L}_{dc} = \sum_{m_1, m_2} \widetilde{\mathcal{W}}(\mu_{m_1}, \mu_{m_2})$;

**Return** $\mathcal{L}_{IID} = \alpha \mathcal{L}_{ic} + \beta \mathcal{L}_{dc} + \sum_{i=1}^{N} \mathcal{L}(f_{\text{IF}}(\boldsymbol{x}^i), y^i)$.

---

### A.3 WASSERSTEIN DISTANCE

Wasserstein distance has its roots in Optimal Transport theory Villani et al. (2009), which is a complete distance metric of distribution. Let $\mu$ be a set of Borel probability measures. Given $\mu_{z^r}, \mu_{z^g} \in \mu$, the corresponding support sets $\sigma_r, \sigma_g$, Wasserstein distance between $\mu_{z^r}$ and $\mu_{z^g}$ is

$$\mathcal{W}(\mu_{z^r}, \mu_{z^g}) = \left( \inf_{\gamma \in \Gamma(x_r, x_g)} \int dis(x_r, x_g)^p d\gamma(x_r, x_g) \right)^{\frac{1}{p}}, \tag{50}$$

where $x_r \in \sigma_r, x_g \in \sigma_g$, $dis(\cdot, \cdot)$ is a distance metric, and $p = 1$ in this paper. $\Gamma(x_r, x_g)$ is the set of all joint distributions $\gamma(x_r, x_g)$ that satisfies $\mu_{z^r} = \int_{x_g} \gamma(x_r, x_g) dx_g$ and $\mu_{z^g} = \int_{x_r} \gamma(x_r, x_g) dx_r$.

### A.4 ALGORITHM

In this subsection, we elaborate on the pseudo-code of proposed IID in Algorithm 1.

### A.5 EXPERIMENTAL SETUP

#### A.5.1 DATASETS

**Vision-language classification.** We execute experiments on four vision-language classification datasets, including Food101 (Wang et al., 2015), MVSA-Single (Niu et al., 2016), MVSA-Multiple (Niu et al., 2016) and HFM (Cai et al., 2019). Food101 comprises images sourced from Google Image Search along with corresponding textual descriptions. MVSA-Single, MVSA-Multiple, and HFM are all derived from Twitter. For Food101, there are 60101 image-text pairs in the training set, 5000 image-text pairs in the validation set, and 21695 image-text pairs in the test set. For MVSA-Single, there are 1555 image-text pairs in the training set. The validation set contains 518 image-text pairs, and the test set consists of 519 image-text pairs. For MVSA-Multiple, there are 17024 image-text pairs, each annotated by three different annotators. The training set contains 13624 image-text pairs, while both the validation set and the test set contain 1700 image-text pairs. For HFM, the training set comprises 19816 image-text pairs, while the validation set contains 2410 image-text pairs, and the test set consists of 2409 image-text pairs.

**Link prediction.** In terms of the link prediction task, we conduct the experiments and evaluate with two standard competition benchmarks, i.e., WN9-IMG Xie et al. (2017) and FB-IMG Sergieh et al. (2018a). M9-IMG dataset is derived from the subset of WN18 Bordes et al. (2013), which embraces structural knowledge as triples, and multimodal knowledge including textual description and visual images. The FB-IMG dataset is derived from the subset of FB15K Mousselly-Sergieh et al. (2018),

which includes structural knowledge consisting of triples extracted from Freebase Bollacker et al. (2008), and multimodal knowledge embracing textual description and visual images.

**Scene recognition.** In accordance with the standard split of the NYU Depth V2 dataset, we consolidate the original 27 categories into 10 categories, encompassing 9 typical scene categories and one "other" category. For the SUN RGB-D dataset, we adhere to the categorization scheme employed by the major baseline methods (QMF (Zhang et al., 2023a) and TMC (Han et al., 2021)), utilizing the 19 primary scene categories, each containing a minimum of 80 images.

### A.5.2 BASELINES

**Baselines of vision-language classification.** To comprehensively evaluate the performance of the proposed IID, both unimodal models and multimodal models are selected as our baselines. Concretely, unimodal models include Bow (Pennington et al., 2014a), Img (Image only, we use ResNet-152 (He et al., 2016) to encode the visual data) and BERT (Devlin et al., 2019a). Multimodal baselines contain Late-fusion, ConcatBow (C-Bow), ConcatBERT (C-BERT), MMBT (Kiela et al., 2019), TMC (Han et al., 2021), DYMM Xue & Marculescu (2023), LCKD Wang et al. (2023), QMF (Zhang et al., 2023a), UniCODE Xia et al. (2023), SimMMDG Dong et al. (2023), and PDF Cao et al. (2024). For Late-fusion and ConcatBERT fusion, we utilize the architecture of ResNet (He et al., 2016) pretrained on ImageNet (Deng et al., 2009) as the backbone network for the visual modality and pretrained BERT (Devlin et al., 2019a) for the text modality. For ConcatBow, we replace BERT with Bow. The Late-fusion conducts an average weighted summarization between visual and textual features, and concat-based fusion concatenates the visual and textual features directly. MMBT leverages the attention mechanism to execute multimodal fusion. TMC proposes a novel trusted multimodal algorithm based on the Dempster-Shafer evidence theory. DYMM employs a gating function to provide modality-level or fusion-level decisions on the fly based on multimodal features QMF designs a robust multimodal fusion method, which is connected to uncertainty learning. PDF derives the multimodal model based on the intra-modal negative and inter-modal positive covariance between the fusion weight and loss function, respectively.

**Baselines of link prediction.** For comprehensive comparison, we select both unimodal methods and multi-modal methods as our benchmark baselines, including TransE Bordes et al. (2013), DistMult Yang et al. (2015), ComplEx Trouillon et al. (2016), RotatE Sun et al. (2019), IKRL Xie et al. (2020), TBKGE Sergieh et al. (2018b), TransAE Wang et al. (2019), MMKRL Lu et al. (2022), and OTKGE Cao et al. (2022).

**Baselines of scene recognition.** For the senses recognition task, we evaluate the proposed methods against various multimodal fusion techniques, including Late-fusion, concatenation-based fusion, align-based fusion (Wang et al., 2016), and the recent state-of-the-art fusion methods, i.e., MMTM (Vaswani et al., 2017)), TMC (Han et al., 2021), and QMF (Zhang et al., 2023a). For Late-fusion and concatenation-based fusion, we employ the ResNet architecture (He et al., 2016), pre-trained on ImageNet, as the backbone network for each modality. Align-based fusion intensifies the similarity of various unimodal features to achieve multimodal alignment.

**Implementation details.** (1) Vision-language classification. In the proposed IID, we employ BERT and ResNet as the latent mappings for text and image modalities, respectively. In the training process, we use BertAdam for the BERT model and regular Adam for the other models. The learning rate is $5e^{-5}$ with a warmup rate of $0.1$. We adopt the early stop strategy based on validation accuracy. We elaborate on the selection of the hyperparameters $\alpha$ and $\beta$ in **Section** A.6. (2) The structured embeddings are produced from triples in knowledge graphs, without any external multi-modal sources. To be specific, unimodal KGE methods such as TransE Bordes et al. (2013) and ComplEx Trouillon et al. (2016) can be used to learn structured embeddings. The linguistic embeddings of entities are learned by adopting the word2vec Mikolov et al. (2013) technique. For instance, we learn the linguistic embeddings of FB-IMG dataset by pre-trained word2vec while we use GloVe Pennington et al. (2014b) for the WN9-IMG dataset. The visual embeddings of entities are learned by pre-trained VGG Simonyan & Zisserman (2015) models. To be specific, visual embeddings are learned by adopting the VGG-m-128CNN Chatfield et al. (2014) model in FB-IMG datasets. As for the WN9-IMG dataset, we take the VGG19 Simonyan & Zisserman (2015) model to learn visual embeddings. (3) Scene recognition. The dimensionalities of unimodal and common representations are set to 128 and 256, respectively. For align-based fusion, we utilize cosine distance to measure the similarity of representations. For the MMTM approach, we adhere to the authors' implementation,

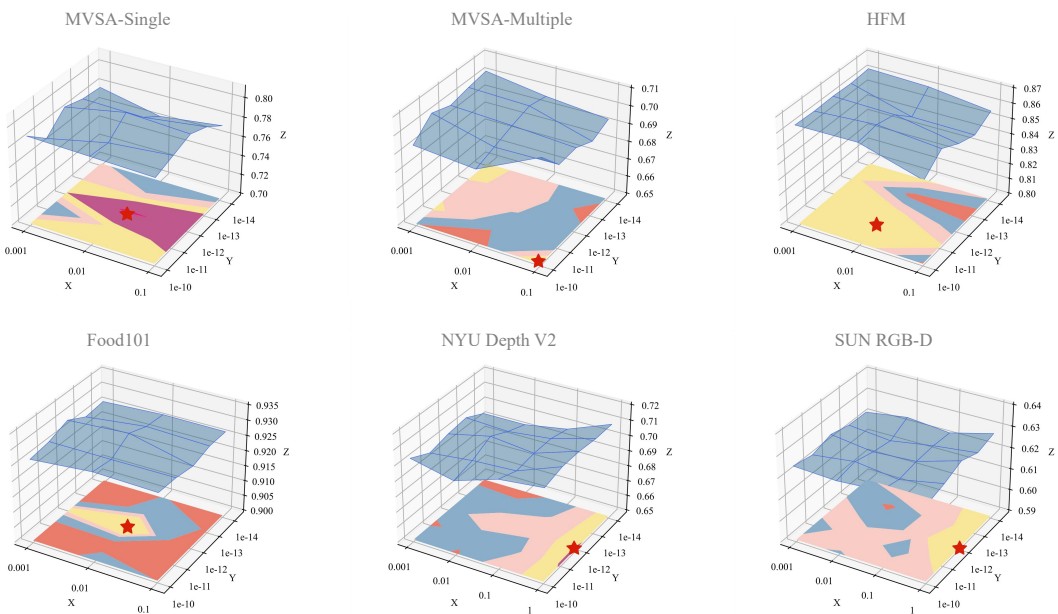

Figure 7: The results of hyperparameters experiments.

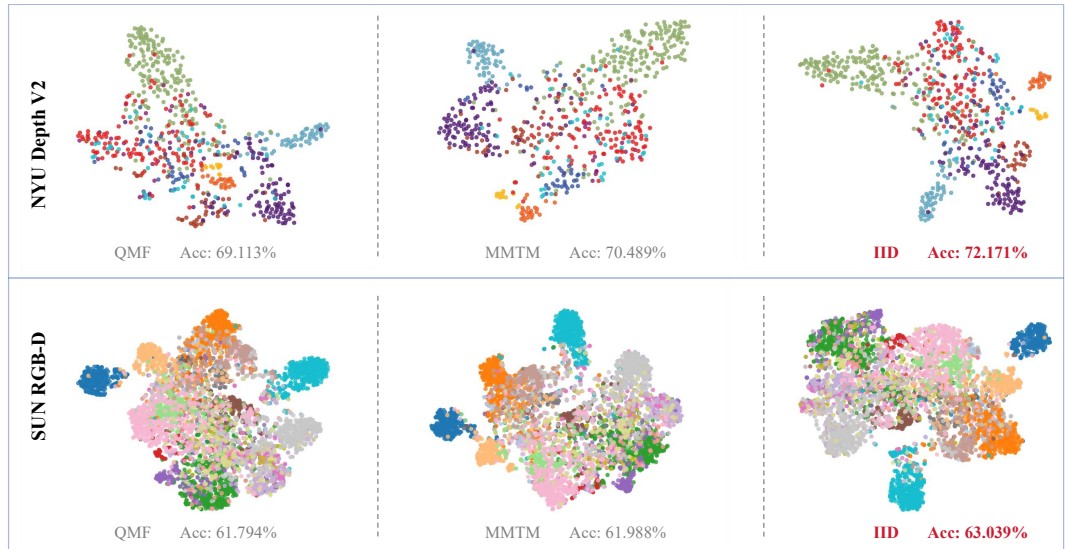

Figure 8: Visualization results of the scene recognition task (the NYU Depth V2 and SUN RGB-D datasets).

setting the squeeze ratio to 4. Across all compared methods, we use the Adam optimizer with $L_2$ regularization and dropout, employing a learning rate of $1 \times 10^{-4}$ and a dropout rate of 0.1.

## A.6 DEEP-GOING EXPERIMENTAL RESULTS

### A.6.1 THE RESEARCH ON THE HYPERPARAMETERS

Two hyper-parameters exist in IID, i.e., $\alpha$ and $\beta$. To understand the impacts of these two hyper-parameters, we conduct empirical comparisons by using various combinations of $\alpha$ and $\beta$ for the proposed IID. As depicted in Equation 13, $\alpha$ controls the impact of informatic constraint on the linear target mapping, and $\beta$ influences the degree of distribution incoherence.

In practice, we search the optimal $\beta$ in $\{1e^{-10}, 1e^{-11}, 1e^{-12}, 1e^{-13}, 1e^{-14}\}$ across all six datasets. As for $\alpha$, on the NYU Depth V2 and SUN RGB-D datasets, we search $\alpha$ in $\{1, 0.1, 0.01, 0.001\}$, while $\alpha$ is searched in $\{0.1, 0.01, 0.001\}$ on other four vision-language classification datasets. We determine the values of $\alpha$ and $\beta$ empirically and depict the results in Figure 7, where the $X$ axis, $Y$ axis, and $Z$ axis represent the value of $\alpha$, the value of $\beta$, and the recognition or classification accuracy, respectively. As we can observe, the optimal combination of $\alpha$ and $\beta$ varies with respect to different datasets, which is indicated by red pentagonal markers. For example, the optimal combinations of $\alpha$ and $\beta$ on the MVSA-Single, MVSA-Multiple and NYU Depth V2 are $\{0.01, 1e^{-12}\}$, $\{0.1, 1e^{-10}\}$, and $\{1, 1e^{-12}\}$, respectively. Therefore, the elaborate assignment of $\alpha$ and $\beta$ can further help to learn informative features, thereby improving the discriminative performance of the proposed method.

### A.6.2 VISUAL COMPARISON

To intuitively demonstrate that IID is capable of learning informative and discriminative representations, we present a visualization of the learned embeddings corresponding to the samples. Specifically, we utilize the $T$-SNE technique (Nkedi-Kizza et al., 2006) to visualize the feature representations of test set samples across multiple datasets (NYU Depth V2, and SUN RGB-D). The visualization results of the scene recognition task (the NYU Depth V2 and SUN RGB-D datasets) are illustrated in Figure 8. We denote the distinct ground truth labels of test set samples by different colors. As we can observe from Figure 8, compared with other multimodal approaches (QMF and MMTM on the NYU Depth V2 and SUN RGB-D datasets), the boundaries of IID between different classes are more distinct, indicating that the IID can better discriminate features across different classes. Additionally, for the proposed IID, data points within the same class tend to cluster more tightly, suggesting that the features extracted by IID have higher intra-class similarity. These observations demonstrate that the IID-learned representations facilitate the extraction of more discriminative features, thereby enhancing performance across various downstream tasks.

### A.6.3 TABLE OF NOTATIONS

We list the definitions of main notations from the main text in Table 5.

Table 5: Main notations used in this paper.

| Notation | Definition |
|---|---|
| **Data and Representation** ||
| $\mathcal{X}, \mathcal{Z}, \mathcal{Y}$ | Input space, latent space, and target space |
| $\mathcal{D}_{\text{train}}, \mathcal{D}_{\text{test}}$ | Training dataset and test dataset |
| $\boldsymbol{x} \in \mathcal{X}$ | Input sample |
| $y \in \mathcal{Y}$ | Label |
| $\boldsymbol{z}_m$ | Representation of modality $m$ |
| $\boldsymbol{z}$ | The fused multimodal feature |
| $M$ | Number of modalities |
| $N$ | The batch size |
| $d$ | The dimension of features |
| **Model Components** ||
| $h(\cdot) : \mathcal{X} \mapsto \mathcal{Z}$ | Latent mapping |
| $g(\cdot) : \mathcal{Z} \mapsto \mathcal{Y}$ | Target mapping |
| $f = gh(\cdot)$ | The composite function of $g(\cdot)$ and $h(\cdot)$ |
| $f_{\text{IF}}$ | Intermediate fusion model |
| $f_{\text{LF}}$ | Late Fusion model |
| $w^m$ | The modality-specific fusion weight |
| $\mathcal{L}(\cdot, \cdot)$ | Cross-Entropy loss function |
| **Theory-related Symbols** ||
| $\boldsymbol{\theta}$ | The parameter of target mapping $g(\cdot)$ |
| $\Lambda$ | The set of parameters |
| $S_m, N_m$ | The index sets of semantic dimensions and noisy dimensions |
| $f^*$ | Bayes optimal hypothesis |
| $\mathscr{G}$ | The generalization error |
| $\mathcal{D}_g$ | The definitional domain of $g$ |
| $\mathcal{D}_{\mathcal{M}}$ | The complete distribution distance metric |
| $\mathcal{H}$ | Hypothesis space |
| $\mu_m$ | The distribution that features of the $m$-th modality are drawn from |
| $\mu$ | The distribution that the multimodal feature $\boldsymbol{z}$ follows |
| $\hat{\mathbb{E}}(f_{\text{IF}})$ | The empirical error of multimodal feature $\boldsymbol{z}$ on $\mathcal{D}_{\text{train}}$ |
| **IID Method** ||
| $\boldsymbol{\theta}^*$ | The theoretically optimal parameter |
| $\hat{\boldsymbol{\theta}}$ | The initial parameter of the linear target mapping |
| $I(\cdot, \cdot)$ | Mutual information computing |
| $\mathcal{L}_{ic}$ | Loss of linear target mapping with informatic constraint |
| $q_{\boldsymbol{\theta}}(\cdot | \cdot)$ | The variational approximation of $p(\cdot | \cdot)$ |
| $\lambda$ | The trade-off hyper-parameter |
| $KL(\cdot)$ | Kullback-Leibler divergence |
| $\mathcal{N}$ | The standard Gaussian distribution |
| $\mathcal{W}(\cdot, \cdot)$ | The analytical form of Wasserstein distance |
| $T$ | The transport plan |
| $C$ | The cost matrix |
| $\mathcal{F}(\cdot)$ | Fast Fourier transform |
| $n \in [1, d]$ | The dimension index |
| $\boldsymbol{\Phi}$ | Gaussian Random matrix |
| $\boldsymbol{\Psi} = \boldsymbol{\Phi}\mathcal{F}^{-1}$ | The RIP-preserved dimensionality reduction matrix |
| $\widetilde{\mathcal{W}}(\cdot, \cdot)$ | The estimation of Wasserstein distance |
| $\mathcal{L}_{dc}$ | Loss of distribution cohering with restricted isometric dimensionality reduction |
| $\alpha, \beta$ | The hyperparameters to control the influence of loss terms |

