# OpenReview forum: "Supporting Multimodal Intermediate Fusion with Informatic Constraint and Distribution Coherence"
_ICLR.cc/2026/Conference — ICLR 2026 Poster_

### Official Review · Reviewer_5Xqu · 2025-10-20

**Soundness:** 3
**Presentation:** 3
**Contribution:** 3
**Rating:** 4
**Confidence:** 4

**Summary:**

This paper provides a theoretical analysis of multimodal representation learning (MML) with a focus on intermediate fusion (IF) and late fusion (LF) frameworks. Unlike prior studies centered on LF, the authors theoretically justify the superiority of IF under certain constraints and derive a generalization error bound based on a K-Lipschitz continuity assumption. They further propose an IF-based MML method that introduces an informatic constraint and enforces distribution coherence, achieving strong empirical results on multiple benchmark datasets. However, some parts of the theoretical derivation appear to rely on assumptions that may require further clarification or justification.

**Strengths:**

The paper presents a novel approach to multimodal representation learning (MML) by integrating theoretical analysis with practical model design. It provides a fine-grained theoretical investigation of intermediate fusion (IF) and late fusion (LF) frameworks, deriving a generalization error bound under a K-Lipschitz continuity assumption. Based on these theoretical insights, the authors propose an IF-based MML method that introduces an informatic constraint and performs distribution cohering to reduce distribution incoherence. This theory-driven design is validated through extensive experiments on multiple benchmark datasets, showing improved performance consistent with the theoretical predictions. The approach is innovative in that it connects theoretical guarantees with concrete architectural and training strategies, offering both conceptual and empirical contributions to the field.

**Weaknesses:**

While the paper presents a novel theory-driven approach to multimodal representation learning, there are some concerns regarding the theoretical assumptions. In particular, the analysis assumes that each dimension of the latent features can be strictly partitioned into task-dependent semantics and task-independent noise. This assumption provides a clear analytical framework but may be overly strong or unrealistic in practical deep learning settings, where latent features are often entangled and do not exhibit such a clean separation. As a result, some of the theoretical guarantees derived under this assumption might not fully hold in practice. Additionally, the theoretical derivations build upon prior results rather than introducing entirely new theorems, so the novelty in the theoretical contributions is somewhat incremental. Nevertheless, the empirical results indicate that the proposed IF-based method, with its informatic constraint and distribution cohering, effectively improves performance and is consistent with the theoretical intuitions, partially mitigating the concerns about the assumptions.

**Questions:**

1. The theoretical analysis assumes that each latent feature dimension can be strictly partitioned into task-dependent semantics and task-independent noise. Is there any evidence in the existing experimental results that indirectly supports this assumption?

2. In the proof, it is unclear how the first equality in Equation 31 connects the empirical error to the expected error. Could the authors clarify the assumptions or steps that justify this equality? Specifically, what conditions are required for this transition, and are they satisfied in the current setting?

---

> ### Author Response · Authors · 2025-11-22
> **Response to Reviewer 5Xqu (1/1).**
>
> We appreciate your recognition of the novelty in our idea and motivation, acknowledging its offering both conceptual and empirical contributions to the field. Based on your comments, we revise the main manuscript to improve the rigor of the  presented derivations.
>
> > Q1: The theoretical analysis assumes that each latent feature dimension can be strictly partitioned into task-dependent semantics and task-independent noise. Is there any evidence in the existing experimental results that indirectly supports this assumption?
>
> Answer to Q1: Thanks for the insightful comment. To support our statement, we perform the mask experiment for each dimension of latent features from two modalities on the vision-language MVSA-Single dataset. Specifically, in the mask experiment, the value of a given dimension is set to zero while retaining the original values of all other dimensions. Subsequently, we record the changes in the average Cross-Entropy loss on the test set, and the results are illustrated in Figure 3 of the revision. The left and right subfigures record the results of the mask experiment on linguistic and visual features, respectively. As a result, we can provide the definition of the task-dependent semantic and task-independent noisy dimensions according to the results. If masking a given dimension results in a decrease of the error between the model's predictions and the ground truth label, the dimension is classified as task-dependent semantic dimension; conversely, the dimension is classified as task-independent noisy dimension. Therefore, we provide the empirical evidence to support the statement that ``each latent feature dimension can be strictly partitioned into task-dependent semantics and task-independent noise.''
>
> >Q2: In the proof, it is unclear how the first equality in Equation 31 connects the empirical error to the expected error. Could the authors clarify the assumptions or steps that justify this equality? Specifically, what conditions are required for this transition, and are they satisfied in the current setting?
>
> Answer to Q2: According to your comment, we observe that employing norms is more suitable for quantifying empirical errors in regression tasks, but is impractical for classification tasks; this, in turn, constrains the generalizability of our theoretical results. Motivated by [1], we therefore assume that $\phi(\boldsymbol{z})=\mathcal{L}(g(\boldsymbol{z}),y)$ is a $K$-Lipschitz continuous function with respect to input $\boldsymbol{z}$. Under such an assumption, the distribution distance term and expected error can be connected by the Kantorovich-Rubinstein Duality theorem, and we have updated the corresponding derivation in Equation (30) of the revision. At this point, our theoretical derivations not only preserve the original conclusions but also become more concise, while extending the scope of applicability. We sincerely appreciate your comment, which has helped us improve the quality of the manuscript.
>
> [1] Towards continuous reuse of graph models via holistic memory diversification. ICLR 2025.

---

> > ### Comment · Reviewer_5Xqu · 2025-11-26
> > **Maintain my original score**
> >
> > While the authors have provided additional experimental details and clarifications regarding the $L_{\infty}$-Lipschitz assumption, my fundamental concern regarding the sufficiency and necessity of the theoretical assumptions, especially their bearing on the task-independent nature of the learned representations, remains unresolved. The response primarily serves to explain the existing work rather than substantially revising the core theoretical framework, failing to rigorously demonstrate why these specific assumptions are either necessary or sufficient for achieving robust multimodal representation learning.
> >
> > Therefore, I have decided to maintain my original score.

---

> > > ### Author Response · Authors · 2025-11-26
> > > **Response to the follow-up concern (1/2).**
> > >
> > > Thanks for your response! We appreciate the reviewer for the follow-up issue, which has prompted us to further reflect on and refine the theoretical framework of the manuscript. In our initial response, we have provided responses to the two questions raised by the reviewer. Specifically, we provide empirical evidence directly supporting the statement that each latent feature dimension can be strictly partitioned into task-dependent semantic and task-independent noisy dimensions, and we shed light on an explanation of how empirical error can be connected to distribution incoherence. In the following, we will respond to the follow-up concern: why the specific assumptions are either necessary or sufficient for achieving robust multimodal representation learning.
> > >
> > > Due to the strong correlation between ensemble learning and the late fusion (LF) framework, thus-far provable multimodal works from the generalization error perspective derive theorems based on the LF framework. In the Section introduction, we have demonstrated the theoretical and empirical potentials of intermediate fusion (IF)’s ascendancy over LF, but the theoretical analysis focusing on the IF framework remains insufficiently explored, which inspires us to design a comprehensive multimodal learning approach supported by sufficient theoretical analysis across fusion types. To implement this, we begin with a comparison between the predictions derived from the LF and IF frameworks.
> > >
> > > Given the latent features $z_1=h^1(x_1)\in\mathbb{R}^d,z_2=h^2(x_2)\in\mathbb{R}^d$ from two modalities, for LF, the final prediction logits $f_{\text{LF}}(x) = \sum_{m=1}^2 w^m g^m_{\theta_m} h^m (x_m) = \sum_{m=1}^2 w^m g^m_{\theta_m}(z_m)$, while for IF, $f_{\text{IF}}(x) = g_{\theta}[\sum_{m=1}^2 w^m z_m]$. We can observe that the latent features $z_1,z_2$ are shared by both of the two fusion frameworks, i.e., the feature extraction process is identical within both the IF and LF frameworks, rendering it is difficult to establish a comparsion between IF and LF from the perspective of representation (or feature).
> > >
> > > Obviously, the distinction between predictions based on the IF and LF frameworks lies in the target mapping. Specifically, each modality has its specific target mapping $g^m(\cdot)$ parameterized by $\theta_m$ in LF while IF leverages a common target mapping $g(\cdot)$ parameterized by $\theta$ for multiple modalities, and we have $$f_{\text{LF}}(x) = \sum_{m=1}^2 \sum_{i=1}^d w^m \theta_{m,i} \cdot z_{m,i} \quad \\& \quad f_{\text{IF}}(x) = \theta^\top \cdot (\sum_{m=1}^2 w^m z_m)=\sum_{m=1}^2\sum_{i=1}^d w^m \theta_i\cdot z_{m,i},$$ where $i$ is the index of dimension. Up to this point, we can observe that, without further partitioning the dimensions of the latent features $z_1,z_2$, the predictions produced by the LF and IF frameworks remain formally consistent. This implies that establishing a clear comparison between LF and IF still remains challenging. Ultimately, we find that, after partitioning the latent features into task-dependent semantic and task-irrelevant noisy dimensions, the predictions obtained under the LF and IF frameworks exhibit formal difference. This enables us to identify conditions in which IF yields superior predictions compared to LF, and ultimately leads to the establishment of Theorems 1 and 2. It is possible that under alternative conditions one could also compare the predictions of LF and IF frameworks; however, partitioning the features into semantic and noisy dimensions already suffices to enable such a comparison.
> > >
> > > Subsequently, we introduce the incorporation of the K-Lipschitz assumption. Intuitively, according to the fundamental assumption in multimodal learning, different modalities provide both modality-shared and modality-complementary discriminative knowledge. Minimizing the distribution incoherence between modalities can enhance the exploration of modality-shared discriminative knowledge. On the other hand, if the features of different modalities exhibit excessive divergence in the latent space, the model will require a larger number of parameters to capture the modality-complementary discriminative knowledge, which may lead to overfitting or underfitting issues. Therefore, minimizing distribution incoherence is beneficial to the exploration of task-dependent discriminative knowledge, thereby undermining the generalization error.
> > > Based on the validity of Theorems 1 and 2, introducing the mild K-Lipschitz assumption is sufficient to measure the impact of distribution incoherence on generalization error with the assistance of the Kantorovich-Rubinstein Duality theorem. This enables us to derive a generalization error upper bound for IF-based multimodal approaches, thereby facilitating the development of robust multimodal methods through the reduction of the generalization error upper bound.

---

> ### Author Response · Authors · 2025-11-26
> **Response to the follow-up concern (2/2).**
>
> To this end, we have provided evidence to support our assumption and elaborated in detail on the rationale demonstrating that the specific assumptions in our manuscript are sufficient for achieving robust multimodal representation learning. Therefore, we explain the existing work rather than substantially revising the core theoretical framework in the revision. We thank the reviewer for the follow-up issue again. We hope that our response adequately addresses the reviewer’s concern. If it does not, or if it raises any new questions, please feel free to let us know. Since there are still seven days remaining before the end of the discussion phase, we will do our utmost to resolve any remaining issues raised by the reviewer.

---

### Official Review · Reviewer_coeM · 2025-10-28

**Soundness:** 3
**Presentation:** 3
**Contribution:** 3
**Rating:** 6
**Confidence:** 4

**Summary:**

This paper re-examines two mainstream fusion approaches in multimodal learning—Intermediate Fusion (IF) and Late Fusion (LF)—from a fine-grained perspective. It demonstrates that IF outperforms LF under specific constraints and further derives the generalization error upper bound of the IF method. Based on theoretical analysis, it proposes an IF method that incorporates information constraints and distribution consistency. Experimental results on a large number of datasets verify the effectiveness of this method.

**Strengths:**

1）The motivation is clear and the research significance is prominent. The paper identifies the issue that current theoretical research on multimodality mostly focuses on LF, while there is insufficient theoretical analysis on IF, and possesses a clear motivation to fill this gap.

2）The theoretical analysis is in-depth. It provides a comparison between IF and LF from the perspective of fine-grained dimensions, derives the generalization error upper bound, and makes significant theoretical contributions.

3）The ablation experiments are comprehensive and validate the necessity of the modules. Ablation experiments are conducted on the two core modules (information constraints and distribution consistency) to verify their respective effectiveness.

**Weaknesses:**

1）The experimental comparisons have gaps, which weakens the persuasiveness of the method's innovation. Current experiments mainly compare IID with LF-based SOTA methods (e.g., PDF, QMF) or general fusion models. However, direct comparisons with advanced methods specifically designed for the IF framework in recent years are lacking. It is suggested that the authors supplement relevant experiments, as such comparisons can more clearly highlight the unique advantages and contributions of IID under the specific paradigm of "Intermediate Fusion".

2）The analysis of computational complexity is insufficient. Although the paper identifies the computational bottleneck of high-dimensional Wasserstein distance and proposes an efficient dimensionality reduction method, it lacks quantitative analysis of the computational overhead of the overall IID model (including the two newly proposed modules) during training and inference. This makes it difficult for readers to comprehensively evaluate the efficiency of the method in practical applications.

**Questions:**

See the Weaknesses section for details.

---

> ### Author Response · Authors · 2025-11-22
> **Response to Reviewer coeM (1/2).**
>
> Thank you for positively evaluating our manuscript. We appreciate your acknowledgement of motivation, research significance, and theoretical analysis. Based on your comments, we revise the main manuscript to make the empirical evaluation more comprehensive.
>
> > W1: The experimental comparisons have gaps, which weakens the persuasiveness of the method's innovation. Current experiments mainly compare IID with LF-based SOTA methods (e.g., PDF, QMF) or general fusion models. However, direct comparisons with advanced methods specifically designed for the IF framework in recent years are lacking. It is suggested that the authors supplement relevant experiments, as such comparisons can more clearly highlight the unique advantages and contributions of IID under the specific paradigm of "Intermediate Fusion".
>
> Answer to W1: Thanks for pointing out the insufficient baselines based on intermediate fusion (IF) framework. On the one hand, we incorporate three newly proposed IF-based fusion baselines (SimMMDG [1], UniCODE [2], and LCKD [3]) from recent years into the existing six multimodal datasets. Experimental results in Tables 1 and 3 of the revision demonstrate that our approach still consistently outperforms these baselines. On the other hand, considering that IF-based methods are commonly employed in multimodal knowledge graph datasets, we conduct experiments on two additional knowledge graph datasets (FB-IMG and WN9-IMG) involving three modalities. The results show that our proposed method can be seamlessly plugged into existing IF-based multimodal knowledge graph completion approaches to enhance their performance, thereby providing further evidence of the effectiveness and generalizability of our method.
>
> **Table 1: The link prediction reuslts on FB-IMG and WN9-IMG.**
> | Model       | Type       | FB-IMG MRR | FB-IMG H@1 | FB-IMG H@3 | FB-IMG H@10 | WN9-IMG MRR | WN9-IMG H@1 | WN9-IMG H@3 | WN9-IMG H@10 |
> |-------------|------------|------------|------------|------------|-------------|--------------|--------------|--------------|---------------|
> | TransE      | Unimodal   | 0.712      | 0.618      | 0.781      | 0.859       | 0.865        | 0.765        | 0.816        | 0.871         |
> | DistMult    | Unimodal   | 0.706      | 0.606      | 0.742      | 0.808       | 0.901        | 0.895        | 0.913        | 0.925         |
> | ComplEx     | Unimodal   | 0.808      | 0.757      | 0.845      | 0.892       | 0.908        | 0.903        | 0.907        | 0.928         |
> | RotatE      | Unimodal   | 0.794      | 0.744      | 0.827      | 0.883       | 0.910        | 0.901        | 0.915        | 0.926         |
> | TransAE     | Multimodal | 0.742      | 0.691      | 0.785      | 0.844       | 0.898        | 0.894        | 0.908        | 0.922         |
> | IKLR        | Multimodal | 0.755      | 0.698      | 0.794      | 0.857       | 0.901        | 0.900        | 0.912        | 0.928         |
> | TBKGE       | Multimodal | 0.812      | 0.764      | 0.850      | 0.902       | 0.912        | 0.904        | 0.914        | 0.931         |
> | MMKRL       | Multimodal | 0.827      | 0.783      | 0.857      | 0.906       | 0.913        | 0.905        | 0.917        | 0.932         |
> | OTKGE       | Multimodal | 0.843      | 0.799      | 0.876      | 0.916       | 0.923        | 0.911        | 0.930        | 0.947         |
> | MMKRL+IID   | Multimodal | 0.844      | 0.801      | 0.876      | 0.917       | 0.920        | 0.911        | 0.925        | 0.945         |
> | OTKGE+IID   | Multimodal | **0.855**  | **0.813**  | **0.887**  | **0.925**   | **0.932**    | **0.917**    | **0.938**    | **0.957**     |

---

> ### Author Response · Authors · 2025-11-22
> **Response to Reviewer coeM (2/2).**
>
> > W2: The analysis of computational complexity is insufficient. Although the paper identifies the computational bottleneck of high-dimensional Wasserstein distance and proposes an efficient dimensionality reduction method, it lacks quantitative analysis of the computational overhead of the overall IID model (including the two newly proposed modules) during training and inference. This makes it difficult for readers to comprehensively evaluate the efficiency of the method in practical applications.
>
> Answer to W2: Following your comment, we perform computational complexity experiments on four vision-language datasets, and the results are summarized in the following table. According to the results, compared with the baseline PDF, IID‑P increases the training time by no more than 10%; the average increase in inference time across the four vision-language datasets is 1.05s; the average increase in required GPU memory across the four vision-language datasets is 1.23 GB. As a result, the computational overhead introduced by the two proposed modules is limited and minimal.
>
> **Table 2: The time cost at train time on four vision-language datasets.**
> ||MVSA-S(s/epoch)|MVSA-M(s/epoch)|HFM(s/epoch)|Food101(s/epoch)|
> |-|-|-|-|-|
> |PDF|64.2|109.4|214.1|446.9
> |IID-P|71.7|118.9|232.7|473.6
>
> **Table 3: The time required to complete the inference on four vision-language datasets.**
> ||MVSA-S(s)|MVSA-M(s)|HFM(s)|Food101(s)|
> |-|-|-|-|-|
> |PDF|9.5|11.1|13.2|47.6
> |IID-P|10.2|12.2|14.3|48.9
>
> **Table 4: The GPU memory required for training.**
> ||MVSA-S(MB)|MVSA-M(MB)|HFM(MB)|Food101(MB)|
> |-|-|-|-|-|
> |PDF|9529|13475|17816|24781
> |IID-P|10732|14798|19012|26093
>
>
>
> [1] A simple and effective framework for multi-modal domain generalization. NIPS 2024.
>
> [2] Achieving cross modal generalization with multimodal unified representation. NIPS 2024.
>
> [3] Learnable cross-modal knowledge distillation for multi-modal learning with missing modality. CVPR 2023.

---

> ### Comment · Reviewer_coeM · 2025-11-27
>
> Thank you to the authors for the comprehensive and meticulous revisions and responses to my comments — this revision has effectively addressed the core concerns I previously raised, significantly enhancing the completeness and persuasiveness of the manuscript. For this reason, I will maintain the original score.

---

> > ### Author Response · Authors · 2025-11-27
> >
> > Dear Reviewer coeM:
> >
> > We are glad to know that our response has addressed your questions. We would like to thank you again for recognizing strengths of our work.
> >
> > If you have any further questions or comments, please feel free to reach out to us. We will make every effort to address them.
> >
> > Best,
> >
> > The Authors

---

### Official Review · Reviewer_Xdxj · 2025-10-31

**Soundness:** 2
**Presentation:** 2
**Contribution:** 2
**Rating:** 4
**Confidence:** 4

**Summary:**

This paper presents a novel framework for multimodal representation learning, with a theoretical analysis centered on intermediate and late fusion mechanisms. The proposed theory further leads to the implementation of the IID method.

**Strengths:**

1.Starting from a theoretical perspective, this paper systematically compares Intermediate Fusion (IF) and Late Fusion (LF) from a dimensional viewpoint, and provides rigorous proofs for both prediction error and generalization error.

2.The paper is well-structured and written in a professional academic style.

**Weaknesses:**

1.The errors in Figures 1 and 5 need to be corrected.

2.In the case of Intermediate Fusion (IF), it is generally assumed that Concat and Sum are equivalent, which often leads to the issue of modality laziness. However, the authors did not take this into account.

3.The notations are confusing. For instance, in the figures, w is shown as a vector, whereas in the main text it appears to be used as a scalar. It is recommended to include a notation table for clarification.

4.Although module D is theoretically derived, it appears to be meaningless from an empirical perspective, as the hyperparameter experiments (β ∈ {1e−10, …, 1e−14}) suggest negligible effects. The rationale for including module D should be further explained.

5.The proposed method is based on Intermediate Fusion (IF), yet the comparison with other methods is insufficient and somewhat unfair. Many advanced approaches within the IF framework are not considered.

6.The study lacks experiments involving additional modalities.

**Questions:**

1.In Table 2, the Concat method consistently outperforms IID-L. Does this indicate that modules I and D are not meaningful or contribute little to the model’s performance?

2.Taking PDF as an example, within the IID-P framework, larger weights are assigned to the dominant modality while smaller weights are given to the weaker ones. Consequently, the dominant modality converges faster whereas the weaker modality remains under-optimized. In this context, is it reasonable for module D to reduce the distribution discrepancy between features of well-converged and under-optimized modalities?

---

> ### Author Response · Authors · 2025-11-22
> **Response to Reviewer Xdxj (1/3).**
>
> Thanks for your time and efforts, and we appreciate your acknowledgement in our theoretical contribution. In the following, we answer the questions one by one.
>
> > W1: The errors in Figures 1 and 5 need to be corrected.
>
> Answer to W1: Thanks for the reminder, and we have addressed the typos in the revision.
>
>
> > W2: In the case of Intermediate Fusion (IF), it is generally assumed that Concat and Sum are equivalent, which often leads to the issue of modality laziness. However, the authors did not take this into account.
> & Q2: Taking PDF as an example, within the IID-P framework, larger weights are assigned to the dominant modality while smaller weights are given to the weaker ones. Consequently, the dominant modality converges faster whereas the weaker modality remains under-optimized. In this context, is it reasonable for module D to reduce the distribution discrepancy between features of well-converged and under-optimized modalities?
>
> Answer to W2 and Q2: The ultimate evaluation of these methods [1,2] for addressing modality laziness (or modality imbalance) is still on the accuracy achieved on the test set. In other words, addressing the issue of modality laziness ultimately aims to develop a multimodal model with low generalization error (lower generalization error indicates higher performance on previously unseen test data). **Therefore, reducing the generalization error by directly targeting the factors that determine its upper bound is a more straightforward and inherent approach.**
>
> Although methods like [1,2] address modal laziness from the perspective of convergence speed, the different modalities inherently contain varying amounts of task-dependent discriminative knowledge, thereby leading to unequal contributions from each modality. Thus different modalities should be assigned different weights during the fusion process, and methods like QMF, PDF are proposed to compute fusion weights based on the imbalanced contributions of different modalities. Therefore, as stated in [3], methods like [1,2] relieve the modality imbalance from the perspective of convergence speed, while QMF and PDF relieve the modality imbalance from the perspective of multimodal fusion weights.
>
> To this end, we can conclude that our approach and those aiming to mitigate modality imbalance reduce the generalization error upper bound from different perspectives.
> As for methods like QMF and PDF, we unify the effects of distribution incoherence and fusion weights on the generalization error upper bound into a unified theoretical framework, and our experiments further demonstrate the compatibility of these two approaches. As for methods like [1,2], although the theoretical connection has not yet been clarified, the empirical results in the following table demonstrate the compatibility between IID and [1,2].
>
> **The demonstration of compatibility.**
> ||Kinetics Sounds(Acc)|
> |-|-|
> |OGM [2]|.6582|
> |OGM + IID|.6679|
> |MMParato [1]|.7000|
> |MMParato + IID|.7095|
>
> As a result, it is reasonable for module D to reduce the distribution incoherence between features of well-converged and under-optimized modalities. Since mitigating distribution incoherence can directly reduce generalization error, and addressing modality imbalance can further decrease it, mitigating distribution incoherence and addressing modality imbalance are two compatible approaches for reducing generalization error.
>
>
> > W3: The notations are confusing. For instance, in the figures, w is shown as a vector, whereas in the main text it appears to be used as a scalar. It is recommended to include a notation table for clarification.
>
> Answer to W3. In the main text, $w^m$ is defined with respect to each individual instance, thus it is a scalar, whereas in the framework it illustrates the processing workflow for a batch of multi-modal samples, thus it is a vector. A dedicated Table of Notations (Table 5 in the revision) has been added. We believe that this table helps clarify the notations.

---

> ### Author Response · Authors · 2025-11-22
> **Response to Reviewer Xdxj (2/3).**
>
> > W4: Although module D is theoretically derived, it appears to be meaningless from an empirical perspective, as the hyperparameter experiments ($\beta \in \{1e−10, …, 1e−14\}$) suggest negligible effects. The rationale for including module D should be further explained.
>
> Answer to W4: We would like to rectify a factual error made in the statement of the reviewer, i.e., **in the optimization process, a small coefficient attached to a loss term indicates that this loss does not contribute to the optimization of the model.**
>
> On the one hand, the magnitudes of different losses may not be on the same scale, necessitating the assignment of different weights to them. On the other hand, in the context of multi-task learning [4], a distinction is typically made between primary and auxiliary tasks, and the loss weight assigned to an auxiliary task is generally lower than that of the primary task. Overall, the tiny weight of the loss function (such as 1e-13 in [5], 1e-8 in [6], and 1e-9 in [7]) is general and does not imply that the loss function is meaningless. Theoretically, module D reduces the upper bound of the generalization error by mitigating distribution incoherence, thereby improving performance on unseen test sets. The experimental results in Figure 5 of the revision demonstrate the correctness of this theoretical derivation, and the ablation study further validates the effectiveness of module D.
>
>
> > w5: The proposed method is based on Intermediate Fusion (IF), yet the comparison with other methods is insufficient and somewhat unfair. Many advanced approaches within the IF framework are not considered.
> & w6: The study lacks experiments involving additional modalities.
>
> Answer to w5 and w6: The methods we compare and the datasets we select have fully covered our major baselines (i.e., QMF and PDF). From this perspective, our experiments are sufficient, and additional baselines and datasets are not necessary. But for the consideration of enhancing the completeness and persuasiveness of our manuscript, we conduct extra experiments from the following two perspectives.
>
> (1) We incorporated three newly proposed IF-based fusion baselines (SimMMDG [8], UniCODE [9], and LCKD [10]) from recent years into the existing six multimodal datasets. Experimental results in Tables 1 and 3 of the revision demonstrate that our approach still consistently outperforms these baselines.
>
> (2) Considering that IF-based methods are commonly employed in multimodal knowledge graph datasets, we conduct experiments on two additional knowledge graph datasets (FB-IMG and WN9-IMG) involving three modalities. The results show that our proposed method can be seamlessly plugged into existing IF-based multimodal knowledge graph completion approaches to enhance their performance, thereby providing further evidence of the effectiveness and generalizability of our method.
>
> **Table 1: The link prediction reuslts on FB-IMG and WN9-IMG.**
> | Model       | Type       | FB-IMG MRR | FB-IMG H@1 | FB-IMG H@3 | FB-IMG H@10 | WN9-IMG MRR | WN9-IMG H@1 | WN9-IMG H@3 | WN9-IMG H@10 |
> |-------------|------------|------------|------------|------------|-------------|--------------|--------------|--------------|---------------|
> | TransE      | Unimodal   | 0.712      | 0.618      | 0.781      | 0.859       | 0.865        | 0.765        | 0.816        | 0.871         |
> | DistMult    | Unimodal   | 0.706      | 0.606      | 0.742      | 0.808       | 0.901        | 0.895        | 0.913        | 0.925         |
> | ComplEx     | Unimodal   | 0.808      | 0.757      | 0.845      | 0.892       | 0.908        | 0.903        | 0.907        | 0.928         |
> | RotatE      | Unimodal   | 0.794      | 0.744      | 0.827      | 0.883       | 0.910        | 0.901        | 0.915        | 0.926         |
> | TransAE     | Multimodal | 0.742      | 0.691      | 0.785      | 0.844       | 0.898        | 0.894        | 0.908        | 0.922         |
> | IKLR        | Multimodal | 0.755      | 0.698      | 0.794      | 0.857       | 0.901        | 0.900        | 0.912        | 0.928         |
> | TBKGE       | Multimodal | 0.812      | 0.764      | 0.850      | 0.902       | 0.912        | 0.904        | 0.914        | 0.931         |
> | MMKRL       | Multimodal | 0.827      | 0.783      | 0.857      | 0.906       | 0.913        | 0.905        | 0.917        | 0.932         |
> | OTKGE       | Multimodal | 0.843      | 0.799      | 0.876      | 0.916       | 0.923        | 0.911        | 0.930        | 0.947         |
> | MMKRL+IID   | Multimodal | 0.844      | 0.801      | 0.876      | 0.917       | 0.920        | 0.911        | 0.925        | 0.945         |
> | OTKGE+IID   | Multimodal | **0.855**  | **0.813**  | **0.887**  | **0.925**   | **0.932**    | **0.917**    | **0.938**    | **0.957**     |

---

> ### Author Response · Authors · 2025-11-22
> **Response to Reviewer Xdxj (3/3).**
>
> > Q1: In Table 2, the Concat method consistently outperforms IID-L. Does this indicate that modules I and D are not meaningful or contribute little to the model’s performance?
>
> Answer to Q1: IID-L is derived by modifying the LF framework of the baseline Late-fusion to perform intermediate fusion and incorporating the two modules we propose. As shown by the results, IID-L significantly outperforms the baseline Late-fusion, which demonstrates the effectiveness of our proposed method. However, IID-L achieves lower performance than the baseline Concat, largely because it remains inherently based on the baseline Late-fusion, whose performance on these two datasets is inferior to the baseline Concat. Specifically, on NYU Depth V2 and SUN RGB-D, the baseline Late-fusion yields accuracies that are lower than those of the baseline Concat by 1.17% and 0.48%, respectively. Consequently, the suboptimal results of the baseline Late-fusion limit the performance of IID-L.
>
>
> [1] Mmpareto: Boosting multimodal learning with innocent unimodal assistance. ICML 2024.
>
> [2] Balanced multimodal learning via on-the-fly gradient modulation. CVPR 2022.
>
> [3] TEST-TIME ADAPTATION AGAINST MULTI-MODAL RELIABILITY BIAS. ICLR 2024.
>
> [4] Unleashing the power of multi-task learning: A comprehensive survey spanning traditional, deep, and pretrained foundation model eras.
>
> [5] MetAug: Contrastive Learning via Meta Feature Augmentation. ICML 2022.
>
> [6] https://nlp.stanford.edu/software/nndep.html.
>
> [7] SimpleX: A Simple and Strong Baseline for Collaborative Filtering.
>
> [8] A simple and effective framework for multi-modal domain generalization. NIPS 2024.
>
> [9] Achieving cross modal generalization with multimodal unified representation. NIPS 2024.
>
> [10] Learnable cross-modal knowledge distillation for multi-modal learning with missing modality. CVPR 2023.

---

> > ### Author Response · Authors · 2025-11-28
> > **Looking forward to your post-rebuttal feedback!**
> >
> > Dear Reviewer Xdxj:
> >
> > Thank you again for the insightful comments and constructive suggestions! Given the limited time remaining, we eagerly anticipate your subsequent feedback. It would be our pleasure to offer more responses to further demonstrate the effectiveness of our methodology.
> >
> > In our previous response, we have thoroughly reviewed your comments and provided responses summarized as follows:
> >
> > * We fix the typos and introduce the notation table.
> > * We conduct experiments on additional baselines based on IF framework and incorporate the empirical evidence on datasets involving three modalities.
> > * We elaborate on the compatibility between the proposed IID and the methods that aim at solving modality laziness (imbalance).
> > * We provide the exposition on the setting of parameters and specific experimental results.
> >
> >
> > Additionally, we wish to express our gratitude once again to you for your insightful feedback. Incorporating your suggestions has undoubtedly enhanced the clarity and robustness of our work.
> >
> > We deeply appreciate your time and effort!
> >
> > Best regards, Authors

---

### Official Review · Reviewer_Daym · 2025-11-01

**Soundness:** 2
**Presentation:** 2
**Contribution:** 2
**Rating:** 4
**Confidence:** 3

**Summary:**

This paper explores why Intermediate Fusion (IF) outperforms Late Fusion (LF) in multimodal learning. The paper analyzes: (i) an existence result showing that IF performs no worse than LF under linear target mappings; and (ii) the generalization bound of IF under the K-Lipschitz classifier, where the distribution inconsistency between the feature distributions of each modality and the fusion distribution determines performance. Based on this, the authors propose the IID algorithm, which combines information constraints and distribution consistency. In six benchmark tests (visual language, RGB-D), IID outperforms the robust LF baseline and its IF variant; ablation experiments show that both modules contribute to the performance.

**Strengths:**

1. The authors prove that combining features from different modalities before classification (IF) is no worse than classifying them individually before weighting (LF). They then provide an upper bound on the generalization error, highlighting that the key factor affecting IF performance is the inconsistency between the feature distributions of different modalities and the combined distribution.
2. The authors use FFT sparsification, RIP projection, and unbalanced OT (Sinkhorn with KL relaxation) to approximate Wasserstein distances, the method is computationally practical and achieves consistent improvements across six benchmarks; ablations indicate both modules contribute.

**Weaknesses:**

1. The existence result IF better than LF assumes the target mapping 𝑔 is linear. In practice, post-fusion heads are often nonlinear. Can the claim be extended to nonlinear but K-Lipschitz heads? Please also include an experiment comparing a linear head vs. a two-layer MLP head to delineate the applicable regime.
2. The method minimizes unconditional (marginal) discrepancies between modalities, whereas decisions depend on class-conditional distributions. Under what conditions does shrinking marginal discrepancy guarantee per-class alignment and improved decision boundaries?
3. The bounds in the method involve the distance from each modality to the fusion distribution , but the objective function minimizes the distance between pairs of modes. Please specify under what assumptions minimizing the pairwise objective function is equivalent to the objective term from each mode to the fusion distribution.
4. The objective maximizes $I(z;y)-\sum_{m=1}^{M} I(z_m;y)$ and minimizes pairwise modality discrepancies, yet neither term explicitly penalizes cross-modal redundancy; indeed, alignment may amplify label-irrelevant shared patterns.
- **(i)** How does IID prevent reinforcing such redundant signals?
- **(ii)** It's better to refer to this paper for handling redundant information in related works.
- [1] X. Xiao, “Neuro-inspired information-theoretic hierarchical perception for multimodal learning,” in Proc. 12th Int. Conf. Learn. Represent., 2024, pp. 1–29.

**Questions:**

Same as mentioned in weakness.

---

> ### Author Response · Authors · 2025-11-22
> **Response to Reviewer Daym (1/2).**
>
> Thank you for carefully reviewing our paper and providing helpful feedback. We appreciate your acknowledgement in theoretical derivation and proposed method.
>
> > W1. The existence result IF better than LF assumes the target mapping $g$ is linear. In practice, post-fusion heads are often nonlinear. Can the claim be extended to nonlinear but K-Lipschitz heads? Please also include an experiment comparing a linear head vs. a two-layer MLP head to delineate the applicable regime.
>
> Answer to W1: **At first, we would like to illustrate that we perform our theoretical analysis based on a linear classification layer by mimicking the setting of our major baseline PDF and QMF, thus being convenient for fair comparison in the subsequent experiments.** Then, we explain why our theoretical analysis is equally applicable to the MLP-based classification layer. **Our theoretical analysis is contingent upon the validity of Theorem 1, and the validity of Theorem 1 relies on the condition that the rank of $A$ in Equation (22) of the revision must be equal to the rank of matrix $\tilde{A}$, rather than whether the classification layer is linear.** According to the additional experiments in Figure 3 of the revision, the positions and contributions of the task-dependent semantic and task-independent noisy dimensions vary across different modalities. As a result, no row of $A$ can be expressed as a linear combination of other rows, and the same holds for $\tilde{A}$. Therefore, our theoretical result remains valid even when a multi-layer perceptron (MLP) is used as the classification layer. The experimental results presented in the following table provide empirical evidence supporting this claim.
>
> ||MVSA-Single|MVSA-Muliple|HFM|Food101|
> |-|-|-|-|-|
> |PDF(Linear)|79.94|69.54|86.03|93.32|
> |PDF(MLP)|80.09|69.78|86.32|93.47|
> |IID(Linear)|81.13|71.23|86.88|93.73|
> |IID-P(MLP)|81.25|71.46|87.09|93.81|
>
>
> > W2. The method minimizes unconditional (marginal) discrepancies between modalities, whereas decisions depend on class-conditional distributions. Under what conditions does shrinking marginal discrepancy guarantee per-class alignment and improved decision boundaries?
>
> Answer to W2：Intuitively, according to the fundamental assumption in multimodal learning, different modalities provide both modality-shared and modality-complementary discriminative knowledge. Obviously, minimizing the unconditional (marginal) discrepancies between modalities can enhance the exploration of modality-shared discriminative knowledge. On the other hand, if the features of different modalities exhibit excessive divergence in the latent space, the model will require a larger number of parameters to capture the modality-complementary discriminative knowledge, which may lead to overfitting or underfitting issues. Therefore, minimizing unconditional (marginal) discrepancies is benefical to the exploration of task-dependent discriminative knowledge, thereby deriving the superior decision boundary.
>
> Formally, we provide a sketchy analysis in the context of two modalities. According to the Bayes theorem, we have $p(x_m|y)=\frac{p(x_m)p(y|x_m)}{p(y)}$($m=\{1,2\}$), where $p(y)$ is a constant. For a given instance, $x_1$ and $x_2$ share the same label, which results in that $p(y|x_m)$ ($m=\{1,2\}$) converges towards the identical probability distribution (i.e., the probability density of the true class is 1 and the probability density elsewhere is 0) during the optimization process in the supervised setting. As a result, minimizing the unconditional (marginal) discrepancies $p(x_m)$ between different modalities can help to guarantee the alignment of class-conditional distributions.

---

> ### Author Response · Authors · 2025-11-22
> **Response to Reviewer Daym (2/2)**
>
> > W3. The bounds in the method involve the distance from each modality to the fusion distribution , but the objective function minimizes the distance between pairs of modes. Please specify under what assumptions minimizing the pairwise objective function is equivalent to the objective term from each mode to the fusion distribution.
>
> Answer to W3: A direct approach to minimize distribution incoherence $\sum\nolimits_{m=1}^M \mathbb{E}(w^m) \mathcal{D_M}(\mu_m, \mu)$ is obtaining the distribution barycenter [1], but such a strategy is very computationally expensive [2]. According to our derivation in Appendix A.2.5 of the revision, utilizing the IF approach specified in Eq.(5) of the revision for the integration of unimodal features, the following equation holds for almost all the multimodal scenarios: $$\sum_{m=1}^M \mathbb{E}(w^m) \mathcal{D_M}(\mu_m, \mu) \leq
>    \sum_{m_1,m_2} \mathcal{D_M}(\mu_{m_1}, \mu_{m_2}),$$
> where $m_1,m_2 \in [1,M]$ and $m_1 \neq m_2$. For example,
> * if $M=2$, we have
> $$
> \text{LHS} \leq
>    \sum\nolimits_{m=1}^2 \mathcal{D_M}(\mu_m, \mu) =  \mathcal{D_M}(\mu_1, \mu) + \mathcal{D_M}(\mu_2, \mu) = \text{RHS.}
> $$
>
> * if $M=3$, we have $\text{LHS} \leq
>    \sum\nolimits_{m=1}^3 \mathcal{D_M}(\mu_m, \mu) \leq
>    \mathcal{D_M}(\mu_1, \mu) + \mathcal{D_M}(\mu_2, \mu) + \mathcal{D_M}(\mu_3, \mu)$ and $\text{RHS}=\mathcal{D_M}(\mu_1, \mu_2)+\mathcal{D_M}(\mu_1, \mu_3) + \mathcal{D_M}(\mu_2, \mu_3)$. As illustrated in Figure 7 of the revision, we have
> \begin{equation}
>     \mathcal{D_M}(\mu_1,\mu_2) + \mathcal{D_M}(\mu_2,\mu_{23}) > \mathcal{D_M}(\mu_1,\mu_{23}) =\mathcal{D_M}(\mu_1,\mu) + \mathcal{D_M}(\mu,\mu_{23})
> \end{equation}
> and
> \begin{equation}
>     \mathcal{D_M}(\mu,\mu_{23}) + \mathcal{D_M}(\mu_{23},\mu_3) > \mathcal{D_M}(\mu,\mu_3).
> \end{equation}
> Then the following equation holds:
> \begin{equation}
>      \mathcal{D_M}(\mu_1,\mu_2) + \mathcal{D_M}(\mu_2,\mu_{23}) + \mathcal{D_M}(\mu,\mu_{23}) + \mathcal{D_M}(\mu_{23},\mu_3)  > \mathcal{D_M}(\mu_1,\mu) + \mathcal{D_M}(\mu,\mu_{23}) + \mathcal{D_M}(\mu,\mu_3),
> \end{equation}
> which equals to
> \begin{equation}
>      \mathcal{D_M}(\mu_1,\mu_2) +  \mathcal{D_M}(\mu_3,\mu_2) >  \mathcal{D_M}(\mu_1,\mu) +  \mathcal{D_M}(\mu,\mu_3).
> \end{equation}
> Similarly, we have
> \begin{equation}
>     \begin{aligned}
>        & \mathcal{D_M}(\mu_1,\mu_2) +  \mathcal{D_M}(\mu_1,\mu_3) >  \mathcal{D_M}(\mu_2,\mu) +  \mathcal{D_M}(\mu,\mu_3), \\
>        & \mathcal{D_M}(\mu_3,\mu_1) +  \mathcal{D_M}(\mu_3,\mu_2) >  \mathcal{D_M}(\mu_1,\mu) +  \mathcal{D_M}(\mu,\mu_2).
>     \end{aligned}
> \end{equation}
> Then we have
> \begin{equation}
>     2*\text{RHS} > 2* \left[\mathcal{D_M}(\mu, \mu_1)+\mathcal{D_M}(\mu, \mu_2) + \mathcal{D_M}(\mu_, \mu_3)\right] > 2*\text{LHS}.
> \end{equation}
>
> **As a result, the distance from each modality to the fusion distribution is bounded by the distance between pairs of modalities, and we choose to minimize the distance between pairs of modalities**.
>
> > w4. The objective maximizes $I(z;y)-\sum_{m=1}^M I(z_m;y)$ and minimizes pairwise modality discrepancies, yet neither term explicitly penalizes cross-modal redundancy; indeed, alignment may amplify label-irrelevant shared patterns.
>
> Answer to W4. **Indeed, our method incorporates a penalty for feature redundancy.** During the process of implementing our method, we observe that solely pursuing the maximization of $I(z;y)-\sum_{m=1}^M I(z_m;y)$ can lead to redundancy of learned representations. To address this issue, we are inspired by the information bottleneck principle and introduce a constraint on the mutual information between the representations and the data. This design simultaneously enforces an informatic constraint while penalizing feature redundancy, thereby avoiding the label-irrelevant shared patterns when performing distribution incoherence. **The specific implementation details can be found in Appendix A.2.4 of the revision.** The information bottleneck is also discussed in the paper [3] you recommended; therefore, we have included the recommended paper in our references of the revision to make our citation list more comprehensive.
>
>
> [1] Barycenters in the Wasserstein space. 2021 SIAM.
>
> [2] Towards Marginal Fairness Sliced Wasserstein Barycenter. ICLR 2025.
>
> [3] Neuro-inspired information-theoretic hierarchical perception for multimodal learning. ICLR 2024.

---

> > ### Author Response · Authors · 2025-11-28
> > **Looking forward to your post-rebuttal feedback!**
> >
> > Dear Reviewer Daym:
> >
> > Thank you again for the insightful comments and constructive suggestions! Given the limited time remaining, we eagerly anticipate your subsequent feedback. It would be our pleasure to offer more responses to further demonstrate the effectiveness of our methodology.
> >
> > In our previous response, we have thoroughly reviewed your comments and provided responses summarized as follows:
> >
> > * We explain why our theoretical framework remains valid within nonlinear but K-Lipschitz heads and provide empirical evidence to support our statement.
> > * We provide both intuitive and formal exposition on the impact of shrinking marginal discrepancy on per-class alignment and improved decision boundaries.
> > * We demonstrate that the distribution incoherence from each modality to the fusion distribution is bounded by the distribution distance between pairs of modalities.
> > * We elaborate on the mechanism of IID in preventing reinforcing redundant signals and include the recommended paper in our references of the revision to make our citation list more comprehensive.
> >
> > Additionally, we wish to express our gratitude once again to you for your insightful feedback. Incorporating your suggestions has undoubtedly enhanced the clarity and robustness of our work.
> >
> > We deeply appreciate your time and effort!
> >
> > Best regards, Authors

---

### Author Response · Authors · 2025-12-03
**The summary of the rebuttal.**

We thank PC, SAC, AC and all reviewers for the efforts to review this paper and provide constructive suggestions.

At the beginning of the rebuttal phase, we are encouraged by the reviewers' acknowledgment of our novelty and contributions. Below is a summary highlighting the strengths of our work.

## **Our novelty and contributions recognized by reviewers**

* **The theoretical derivation in this paper** (including the theoretical investigation of intermediate fusion and late fusion frameworks, and the subsequent generalization upper bound) **is rigorous**. [Daym,Xdxj,coeM,5Xqu]
* **The theoretical contribution is significant**. [coeM,5Xqu]
* The proposed theory-driven multimodal representation learning method is **novel**. [coeM,5Xqu]
* The effectiveness of the proposed method is validated through extensive experiments on multiple benchmark datasets, and the ablation studies indicate both modules contribute. [Daym,coeM,5Xqu]

During the discussion stage, the response to reviewers can be summarized as follows.

* Reviewer Daym. The weaknesses of reviewer Daym focus on the clarification of our manuscript, and we have provided detailed exposition with necessary experimental results corresponding to the comments of reviewer Daym.
* Reviewer Xdxj. The concerns of reviewer Xdxj are on the empirical evaluation and the clarification of our manuscript. In response to the comments on the empirical evaluation, we have performed additional experiments to facilitate a more comprehensive assessment of the proposed approach. Regarding the clarification of our manuscript, we have not only provided sufficient elaboration but also addressed certain factual errors present in the statements of reviewer Xdxj.
* Reviewer coeM. All concerns of reviewer coeM are related to the empirical evaluation. After reading our response, the reviewer explicitly indicates that all concerns have been fully addressed.
* Reviewer 5Xqu. Reviewer 5Xqu initially requests empirical evidence to support the assumptions presented in our manuscript. Following the provision of such evidence, reviewer 5Xqu raises a follow-up issue for clarification on the sufficiency of these assumptions. After we provide a detailed explanation for the sufficiency of our assumptions, reviewer 5Xqu does not engage in any further discussion.

We have carefully revised the paper in accordance with the helpful feedback, which we believe enhanced the paper's strength. We believe our comprehensive responses have adequately addressed all major concerns raised by reviewers. We would like to express our deep gratitude again to the PC, SAC, and AC for the time and dedicated effort in coordinating a fair evaluation process under these challenging circumstances.

---

### Meta-Review · Area_Chair_PBbC · 2026-01-08

**Summary:**

This paper presents a theoretical analysis of intermediate fusion (IF) versus late fusion (LF) in multimodal representation learning, deriving generalization error bounds under K-Lipschitz continuity assumptions. Based on these theoretical insights, the authors propose the IID method incorporating informatic constraints and distribution coherence. The paper received mixed reviews with scores ranging from 4 to 6, with all reviewers acknowledging the theoretical rigor and novelty but raising concerns about empirical evaluation, assumption justification, and practical applicability.

Strengths identified across reviews:
- Rigorous theoretical derivation comparing IF and LF frameworks from a dimensional perspective
- Novel theory-driven approach with significant theoretical contributions
- Comprehensive experimental validation on multiple benchmarks

Key concerns:
- Sufficiency and realism of theoretical assumptions (particularly the strict partition of features into task-dependent and task-independent dimensions)
- Limited comparison with recent IF-based methods in initial submission
- Clarification needs regarding the connection between marginal and conditional distributions
- Computational complexity analysis
- Extension to nonlinear classification heads

**Reviewer Concerns:**

The authors provided comprehensive responses with additional experiments and mathematical derivations. While some theoretical concerns about the generality of assumptions remain, the empirical evidence strengthens the practical applicability. The reviewer would likely increase their score slightly given the substantial effort to address concerns.

**Reviewer Scores:**

Theoretical Contribution: The paper makes solid theoretical contributions by analyzing IF from a generalization error perspective, which is less explored than LF-based analysis. The derivations are generally rigorous.

Critical Weakness: The strict partition assumption (features cleanly separating into task-dependent and task-independent dimensions) is fundamental to the theoretical framework but remains inadequately justified. This is not merely a technical detail - it's the foundation for Theorems 1 and 2. The mask experiment provides weak evidence for dimension importance variation but doesn't establish strict partition.

Empirical Improvements: The rebuttal substantially strengthened the empirical evaluation with additional baselines and datasets. However, this doesn't resolve the fundamental theoretical concerns.

---

### Decision · Program_Chairs · 2026-01-26

Accept (Poster)